# Promoter-anchored chromatin interactions predicted from genetic analysis of epigenomic data

Yang Wu [1,7], Ting Qi [1,7], Huanwei Wang [1], Futao Zhang[1], Zhili Zheng[1,2], Jennifer E. Phillips-Cremins [3], Ian J. Deary[4,5], Allan F. McRae [1], Naomi R. Wray [1,6], Jian Zeng[1] & Jian Yang [1,2 ✉]

Promoter-anchored chromatin interactions (PAIs) play a pivotal role in transcriptional regulation. Current high-throughput technologies for detecting PAIs, such as promoter capture Hi-C, are not scalable to large cohorts. Here, we present an analytical approach that uses summary-level data from cohort-based DNA methylation (DNAm) quantitative trait locus (mQTL) studies to predict PAIs. Using mQTL data from human peripheral blood ($n = 1980$), we predict 34,797 PAIs which show strong overlap with the chromatin contacts identified by previous experimental assays. The promoter-interacting DNAm sites are enriched in enhancers or near expression QTLs. Genes whose promoters are involved in PAIs are more actively expressed, and gene pairs with promoter-promoter interactions are enriched for co-expression. Integration of the predicted PAIs with GWAS data highlight interactions among 601 DNAm sites associated with 15 complex traits. This study demonstrates the use of mQTL data to predict PAIs and provides insights into the role of PAIs in complex trait variation.

[1] Institute for Molecular Bioscience, The University of Queensland, Brisbane, QLD 4072, Australia. [2] Institute for Advanced Research, Wenzhou Medical University, 325027 Wenzhou, Zhejiang, China. [3] Department of Bioengineering, University of Pennsylvania, Philadelphia, PA 19104, USA. [4] Centre for Cognitive Ageing and Cognitive Epidemiology, University of Edinburgh, Edinburgh EH8 9JZ, UK. [5] Department of Psychology, University of Edinburgh, Edinburgh EH8 9JZ, UK. [6] Queensland Brain Institute, The University of Queensland, Brisbane, QLD 4072, Australia. [7] These authors contributed equally: Yang Wu, Ting Qi. ✉email: jian.yang.qt@gmail.com

Genome-wide association studies (GWASs) in the past decade have identified tens of thousands of genetic variants associated with human complex traits (including common diseases) at a stringent genome-wide significance level[1,2]. However, most of the trait-associated variants are located in non-coding regions[3,4], and the causal variants as well as their functional roles in trait aetiology are largely unknown. One hypothesis is that the genetic variants affect the trait through genetic regulation of gene expression[4]. Promoter-anchored chromatin interaction (PAI)[5,6] is a key regulatory mechanism whereby non-coding genetic variants alter the activity of cis-regulatory elements and subsequently regulate the expression levels of the target genes. Therefore, a genome-wide map of PAIs is essential to understand transcriptional regulation and the genetic regulatory mechanisms underpinning complex trait variation.

High-throughput experiments, such as Hi-C[7] and ChIA-PET (chromatin interaction analysis by paired-end tag sequencing)[8], have been developed to detect chromatin interactions by a massively parallelised assay of ligated DNA fragments. Hi-C is a technique based on chromosome conformation capture (3C)[9] to quantify genome-wide interactions between genomic loci that are close in three-dimensional (3D) space, and ChIA-PET is a method that combines ChIP-based methods[10] and 3C. However, these high-throughput assays are currently not scalable to population-based cohorts with large sample sizes because of the complexity of generating a DNA library and the extremely high-sequencing depth needed to achieve high detection resolution[11]. On the other hand, recent technological advances have facilitated the use of epigenomic marks to infer the chromatin state of a specific genomic locus and further to predict the transcriptional activity of a particular gene[12,13]. There have been increasing interests in the use of epigenomic data (e.g., DNA methylation (DNAm) and/or histone modification) to infer chromatin interactions[14–17]. These analyses, however, rely on individual-level chromatin accessibility data often only available in small samples[14,16], and it is not straightforward to use the predicted chromatin interactions to interpret the variant-trait associations identified by GWAS.

In this study, we propose an analytical approach to predict chromatin interaction by detecting the association between DNAm levels of two CpG sites due to the same set of genetic variants (i.e., pleiotropic association between DNAm sites). This can be achieved because if the methylation levels of a pair of relatively distal CpG sites covary across individuals and such covariation is partly caused by a set of shared genetic variants in cis (Fig. 1b), it is very likely that the two genomic regions interact (having contacts or functional links because of their close physical proximity in 3D space). Our analytical approach is based on two recently developed methods, i.e., the summary-data–based Mendelian randomisation (SMR) test and the test for heterogeneity in dependent instruments (HEIDI)[18], which are often used in combination to detect pleiotropic association between a molecular phenotype (e.g., gene expression or DNA methylation) and a complex trait[18] or between two molecular phenotypes[19]. The SMR & HEIDI approach only requires summary-level data from DNA methylation quantitative trait locus (mQTL) studies, providing the flexibility of using mQTL data from studies with large sample sizes to ensure sufficient power. Since the proposed method is based on cohort-based genetic data, it also allows us to integrate the predicted chromatin interactions with GWAS results to understand the genetic regulatory mechanisms for complex traits. In this study, we analyse mQTL summary data from a meta-analysis of two cohort-based studies on 1980 individuals with DNAm levels measured by Illumina 450K methylation arrays and SNP data from SNP-array-based genotyping followed

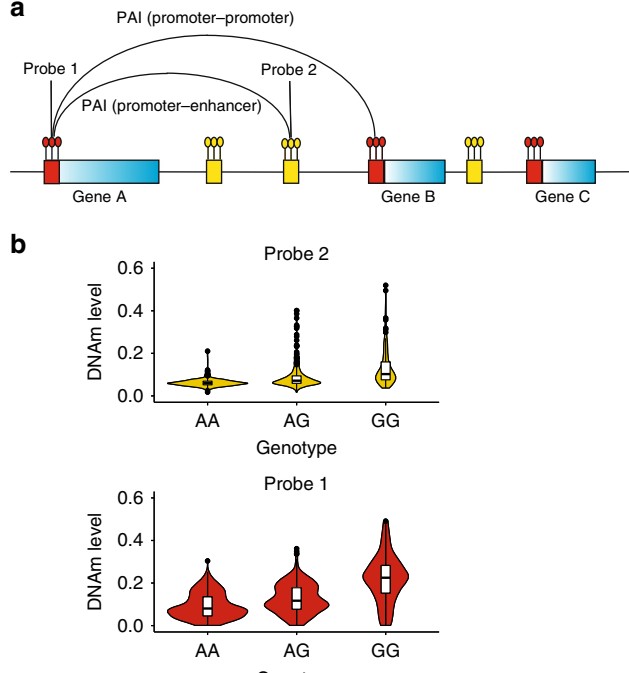

**Fig. 1 Schematic of the promoter-anchored chromatin interaction (PAI) analysis. a** A schematic of the PAI analysis. The blue rectangles represent genes with their promoter regions colour coded in red. The small yellow bars represent other functional regions (e.g., enhancers). In this toy example, the promoter region of Gene A is used as the bait for the PAI analysis. Genes (e.g., genes A and B) whose promoters are involved in significant PAIs are defined as Pm-PAI genes. DNAm sites (e.g., DNAm probe 2) that showed significant interactions with the DNAm sites in promoter regions are defined as promoter-interacting DNAm sites or PIDS. **b** DNAm levels of two CpG sites are associated because of shared causal variant(s). The DNAm level ranges from 0 to 1 (with 0 being unmethylated and 1 being fully methylated). It is the ratio of the methylated probe intensity to the overall intensity (sum of methylated and unmethylated probe intensities). On the violin plots, the centre line shows the median, box limits are the upper and lower quartiles, whiskers represent 1.5× interquartile range and individual points are outliers.

by imputation to the 1000 Genome Project (1KGP) reference panels[19,20].

## Results

**Predicting PAIs using mQTL data**. As described above, our underlying hypothesis was that if the DNAm levels of two relatively distal CpG sites are associated due to the same set of causal genetic variants (Fig. 1b), then it is very likely that these two chromatin regions have contacts or functional links because of their close physical proximity in 3D space. Hence, we set out to predict PAIs from mQTL data. We applied the SMR & HEIDI approach[18] to test for pleiotropic associations of a DNAm site in the promoter region of a gene with all the other DNAm sites within 2 Mb distance of the focal promoter in either direction (excluding those in the focal promoter) using mQTL summary data from peripheral blood samples (Fig. 1, Supplementary Fig. 1 and Methods). The mQTL summary data were generated from a meta-analysis of two mQTL data sets from McRae et al. ($n = 1980$)[19,20]. After quality controls (Methods), there were 90,749 DNAm probes with at least one cis-mQTL with $P_{mQTL} < 5 \times 10^{-8}$, 28,732 of which were located in promoters annotated based on data from blood samples of the Roadmap Epigenomics Mapping Consortium (REMC)[13]. In total, we

identified 34,797 PAIs between pairwise DNAm sites that passed the SMR test ($P_{SMR} < 1.76 \times 10^{-9}$ based on a Bonferroni correction for multiple tests; the SMR $P$-values were computed based on the two-sided Wald test[18]) and were not rejected by the HEIDI test ($P_{HEIDI} > 0.01$; see Wu et al.[19] for the justification of the use of this HEIDI threshold $P$-value; the HEIDI $P$-values were computed from an approximate, one-sided, sum of chi-squared test[18]). The significant PAIs comprises 21,787 unique DNAm sites, among which 10,249 were the exposure probes in promoter regions of 4617 annotated genes. Most of the DNAm sites in promoters showed pleiotropic associations with multiple DNAm sites (mean = 4) (Supplementary Fig. 2a). The distances between 95% of the pairwise interacting DNAm sites were less than 500 Kb (mean = 79 Kb and median = 23 Kb). Only ~0.7% of the predicted PAIs were between DNAm sites greater than 1 Mb apart (Supplementary Fig. 2b). The summary statistics of the predicted PAIs are publicly available through the M2Mdb Shiny online application (see http://cnsgenomics.com/shiny/M2Mdb/).

**Overlap of the predicted PAIs with chromatin contacts**. We first examined whether the predicted PAIs are consistent with chromatin contacts identified by experimental assays, such as Hi-C[21] and promoter captured Hi-C (PCHi-C)[5]. While the majority of experimental assays are measured in primary cell lines, topological-associated domains (TADs) annotated from Hi-C are relatively conserved across cell types[22]. We therefore tested the overlap of our predicted PAIs with the TADs identified from recent Hi-C and PCHi-C studies[5,21,23] (see Supplementary Table 1 for a full list of data sets used in this study). We found that 22,024 (63.3%) of the predicted PAIs were between DNAm sites located in the TADs identified by Rao et al.[21] using Hi-C in the GM12878 cell lines, 27,200 (78.2%) in those by Dixon et al.[23] using Hi-C in embryonic stem cells, and 27,716 (79.7%) in those by Javierre et al.[5] using PCHi-C in primary hematopoietic cells[5], all of which were significantly higher than expected by chance (empirical $P_{enrichment} < 0.001$; Fig. 2a–c). Note that the $P$-value was computed by comparing the observed number to a null distribution generated by resampling the same number of DNAm pairs at random from distance-matched DNAm pairs included in the SMR analysis (Methods); the $P$-value was truncated at 0.001 due to the finite number of resampling. One example was the *MAD1L1* locus (a ~450 Kb region) on chromosome 7 (Fig. 2d, e) where there were a large number of predicted PAIs highly consistent with TADs identified by Hi-C from the Rao et al.[21] study. There were also scenarios where the predicted PAIs were not aligned well with the TAD data. For example, 107 of the 183 predicted PAIs at the *RPS6KA2* gene locus did not overlap with the TADs identified by Hi-C from the Rao et al. study[21] (Supplementary Fig. 3a). These predicted interactions, however, are very likely to be functional as indicated by our subsequent analysis with GWAS and omics data (see below). Additionally, the predicted PAIs were slightly enriched for the Hi-C loops identified from Rao et al.[21] (1.49-fold, empirical $P_{enrichment} < 0.001$, $m$ = 130; Fig. 3a) and the *POLR2A* ChIA-PET loops from the ENCODE[24] project (1.44-fold, empirical $P_{enrichment} < 0.001$, $m$ = 2315; Fig. 3b), although the numbers of overlaps were small. One notable example was the *GNB1* locus where the predicted PAI between the promoter region of *GNB1* and an enhancer nearby is consistent with the enhancer-promoter interaction identified by both Hi-C from Rao et al.[21] and PCHi-C from Jung et al.[25] in the GM12878 cell lines (Supplementary Fig. 4).

**Comparison with other prediction methods**. To assess the performance of our PAI prediction method, we compared it with two state-of-the-art approaches of this kind, i.e., the correlation-

based method used in Gate et al.[26] and the pairwise hierarchical model (PHM) method developed by Kumasaka et al.[17], using the DNAm data described above or the chromatin accessibility data (measured by Assay for Transposase-Accessible Chromatin using sequencing (ATAC-seq)) from Kumasaka et al.[17]. We used a recently released chromatin interaction data (PCHi-C loops) generated by Jung et al.[25] in GM12878 cell lines for validation, and quantified the enrichment of the predicted interactions in the PCHi-C loops defined based on a range of PCHi-C $P$-value thresholds (Methods). We chose the PCHi-C data from Jung et al. because the $P$-values of all the tested loops are available and because compared to other Hi-C data sets, chromatin interactions identified in GM12878 cell lines may be more relevant to the predicted PAIs in whole blood. The results showed that our predicted PAIs using either DNAm or chromatin accessibility data were highly enriched in the PCHi-C loops and that the fold enrichment increased with the increase of the significance level used to claim the PCHi-C loops (Fig. 3c), consistent with the observation from previous work that Hi-C loops with lower $P$-values are more reproducible between biological replicates[27]. Our SMR & HEIDI method outperformed the correlation-based method using either DNAm or chromatin accessibility data, as evidenced by the larger fold enrichment of our method compared to the correlation-based method at all the PCHi-C significance levels (Fig. 3c). We also compared the predicted PAIs with the interactions identified from the PHM approach[17] using the chromatin accessibility data. Of the 15,487 interactions identified by the PHM approach, 10,416 were tested in our SMR & HEIDI analysis; 98.4% were replicated at a nominal significance level ($P_{SMR} < 0.05$ and $P_{HEIDI} > 0.01$), and 36% were significant after multiple testing correction ($P_{SMR} < 4.8 \times 10^{-6}$ (0.05/10, 416) and $P_{HEIDI} > 0.01$). While the PHM approach requires individual-level genotype and chromatin accessibility data and is less computationally efficient due to the use of Bayesian hierarchical model, our SMR & HEIDI method that requires only summary-level data is more flexible and can be potentially applied to all epigenetic QTL data.

We further performed an aggregate peak analysis (APA) implemented in Juicer[28] to evaluate the performance of the methods by the aggregate enrichment of the predicted interactions in the combined Hi-C map in GM12878 produced by Rao et al.[21]. We observed that the PAIs predicted by SMR & HEIDI from DNAm data showed the strongest enrichment among all the analyses although the APA score of PHM was higher than that of SMR & HEIDI when using chromatin accessibility data (Supplementary Table 2). In addition, Juicer APA also reported a Peak to Mean (P2M) value for each Hi-C loop (or predicted interaction) to indicate its enrichment compared to nearby regions in a Hi-C map. We observed that the P2M value was neither correlated with the strength of a Hi-C loop nor with that of a predicted interaction (Supplementary Table 3).

**Enrichment of the predicted PAIs in functional annotations**. To investigate the functional role of the DNAm sites that showed significant interactions with the DNAm sites in promoter regions (called promoter-interacting DNAm sites or PIDSs hereafter), we conducted an enrichment analysis of the PIDSs ($m$ = 14,361) in 14 main functional annotation categories derived from the REMC blood samples (Methods). The fold enrichment was computed as the proportion of PIDSs in a functional category divided by the mean of a null distribution generated by resampling variance-matched "control" probes at random from all the outcome probes used in the SMR analysis. We found a significant enrichment of PIDSs in enhancers (fold-enrichment = 2.17 and empirical $P_{enrichment} < 0.001$), repressed

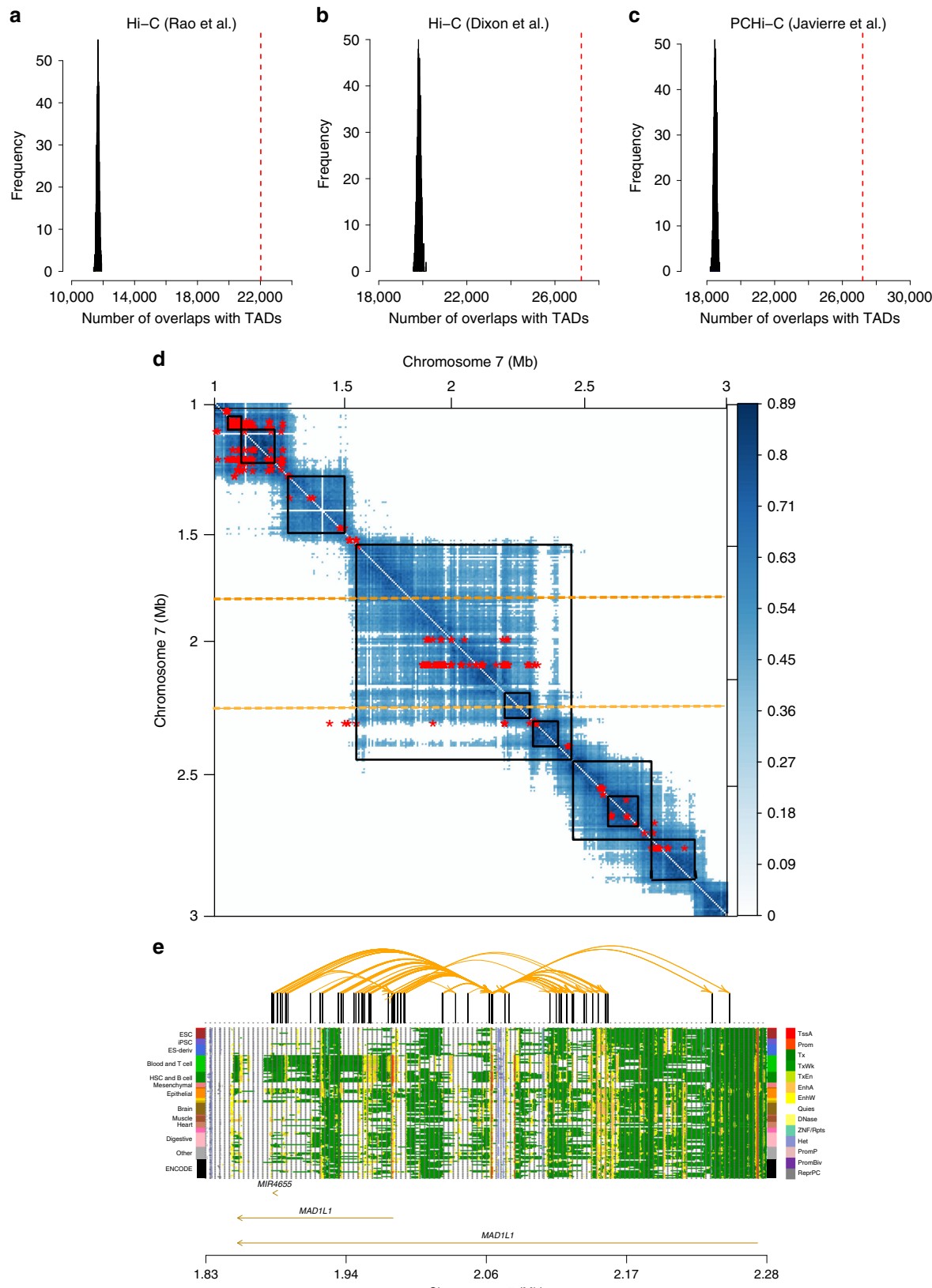

Polycomb regions (fold-enrichment = 1.56 and empirical $P_{enrichment}$ < 0.001), primary DNase (fold-enrichment = 1.43 and empirical $P_{enrichment}$ < 0.001) and bivalent promoters (fold-enrichment = 1.12 and empirical $P_{enrichment}$ < 0.001) and a significant under-representation in transcription starting sites (fold-enrichment = 0.21 and empirical $P_{enrichment}$ < 0.001), quiescent regions (fold-enrichment = 0.74 and empirical $P_{enrichment}$ < 0.001), promoters around transcription starting sites (fold-enrichment = 0.77 and empirical $P_{enrichment}$ < 0.001), and transcribed regions (fold-enrichment = 0.90 and empirical

**Fig. 2 Overlap of the predicted PAIs with TADs identified by Hi-C and PCHi-C. a**, **b**, **c** overlaps of the predicted PAIs with TADs identified by **a** Rao et al.[21] and **b** Dixon et al.[23] using Hi-C and by **c** Javierre et al.[5] using PCHi-C. The red dash lines represent the observed number and histograms represent the distribution of control sets. **d** A heatmap of the predicted PAIs (red asterisks) and chromatin interactions with correlation scores >0.4 (blue dots) identified by Grubert et al.[57] using Hi-C in a 2 Mb region on chromosome 7. Black squares represent the TADs identified by Rao et al.[21]. The heatmap is asymmetric for the PAIs (red asterisks) with the x- and y-axes representing the physical positions of outcome and exposure DNAm probes, respectively. **e** the predicted PAIs at the *MAD1L1* locus, a 450-Kb sub-region of that shown between two orange dashed lines in **d**. The orange curved lines on the top represent the significant PAIs between 14 DNAm sites in the promoter regions of *MAD1L1* (multiple transcripts) and other DNAm sites nearby. The panel on the bottom represents 14 chromatin state annotations (indicated by different colours) inferred from data of 127 REMC samples (one row per sample). Note that the predicted PAIs appear to be much sparser than the Hi-C loops largely because the PAIs were predicted from analyses with very stringent significance levels (see Supplementary Note 2 for discussion).

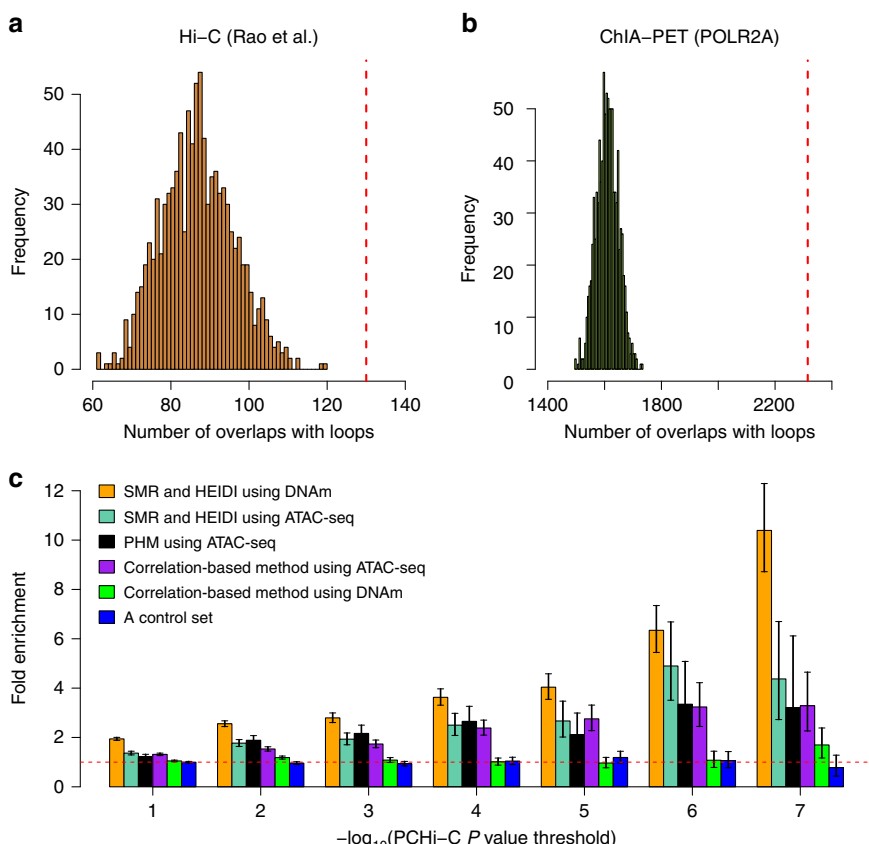

**Fig. 3 Enrichment of the predicted interactions in chromatin loops. a**, **b** Overlaps of the predicted PAIs with the chromatin loops identified by **a** Hi-C from Rao et al.[21] and **b** *POLR2A* ChIA-PET from the ENCODE project[24]. The red dash lines represent the observed number and histograms represent the distribution of control sets. **c** enrichment of the predicted interactions in the significant PCHi-C loops defined based on a range of *P*-value thresholds (*P*-values were obtained from Jung et al.[25]). We used the PCHi-C loops identified from Jung et al. in GM12878 cell lines[25]. PHM: the pairwise hierarchical model developed by Kumasaka et al.[17]. The fold enrichment value was computed by a 2 × 2 contingency table and the Fisher's exact test was used to assess the statistical significance of the enrichment. The error bar around each estimate represents the 95% confidence interval. The horizontal red dashed line indicates no enrichment.

$P_{enrichment} < 0.001$) (Fig. 4a, b). On one hand, the enrichment test is not biased by the fact that the Illumina 450K methylation array probes are preferentially distributed towards certain genomic regions (e.g., promoters; Fig. 4a) because it tests against control probes sampled from probes on the array rather than random genomic positions. On the other hand, however, this test is over conservative because the control probes are enriched in certain functional genomic regions (Supplementary Fig. 5a) and can possible contain some of the PIDSs, which may explain the relatively small fold enrichments observed above. When we tested the enrichment against random genomic positions, the fold enrichment values were several-fold larger than those computed against array probes (Supplementary Fig. 5b). The depletion of PIDSs in promoters was due to the exclusion of outcome probes from the focal promoters (Methods;

Supplementary Fig. 6). In addition, a large proportion (~18%) of the predicted PAIs were promoter-promoter interactions (PmPmI), consistent with the results from previous studies[5,29] that PmPmI were widespread.

We also examined whether our predicted PAIs were enriched in the binding regions of proteins known to be involved in 3D organisation of the genome. We used the chromatin immuno-precipitation sequencing (ChIP-Seq) data from GM12878 for four DNA-binding proteins (i.e., CTCF, Rad21, ZNF143, YY1) from the ENCODE project[24] (Methods). Of the 21,787 unique DNAm sites that showed significant PAIs, 9454 (43.4%), 7588 (34.8%), 6854 (31.5%) and 9477 (43.5%) were located in the binding regions of CTCF, Rad21, ZNF143 and YY1, respectively. These overlaps were significantly larger than those for a random set of

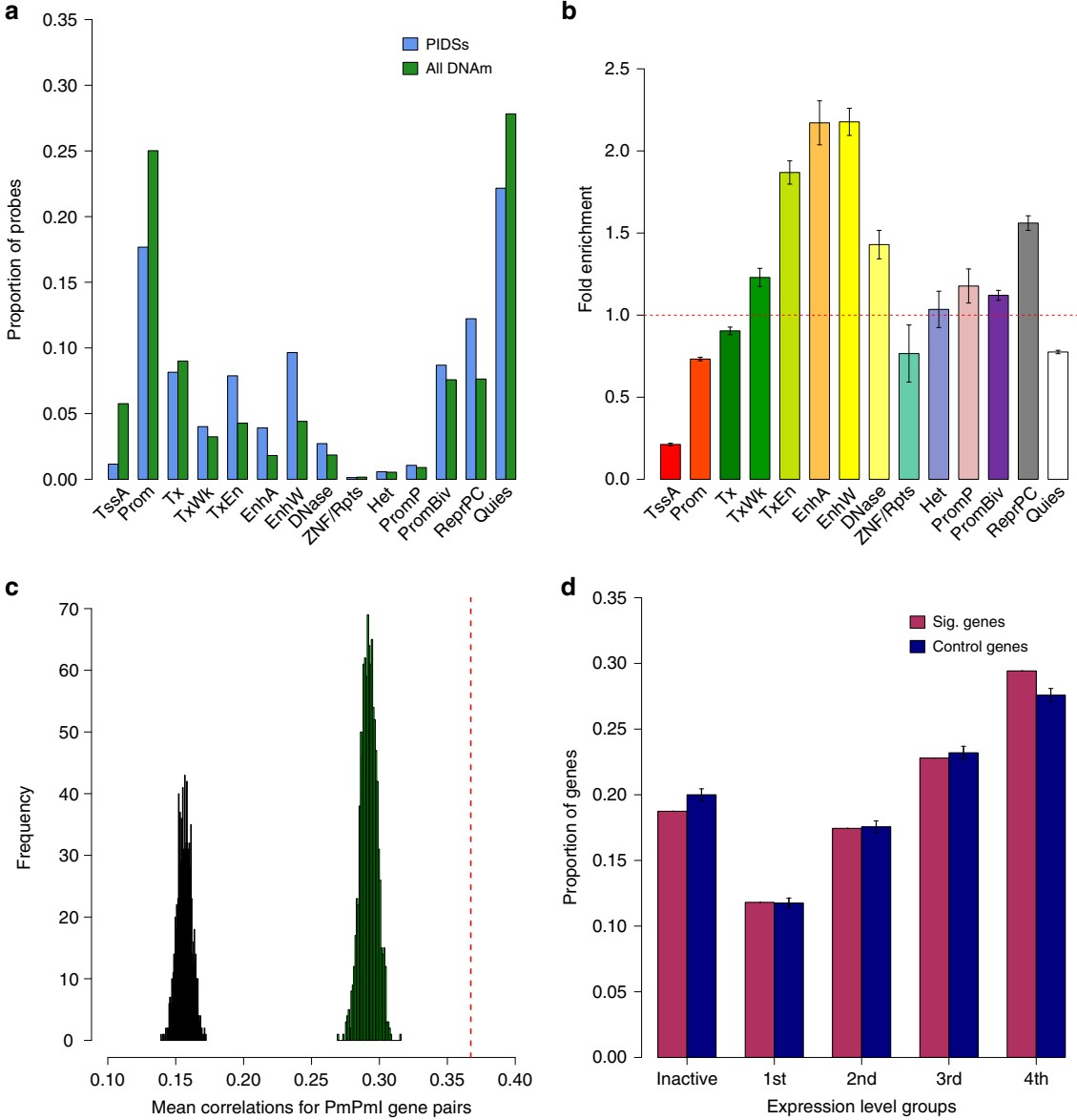

**Fig. 4 Enrichment of PIDSs and Pm-PAI genes. a, b** enrichment of PIDSs in 14 main functional annotation categories inferred from the 127 REMC samples. Fold enrichment: a ratio of the proportion of PIDSs in an annotation category to the mean of the control sets. Data are presented as mean $+/-$ standard deviation (SD). The error bar in **b** represents SD of the estimate under the null obtained from the 1000 control sets. The 14 functional categories are: TssA active transcription start site, Prom upstream/downstream TSS promoter, Tx actively transcribed state, TxWk weak transcription, TxEn transcribed and regulatory Prom/Enh, EnhA active enhancer, EnhW weak enhancer, DNase primary DNase, ZNF/Rpts state associated with zinc finger protein genes, Het constitutive heterochromatin, PromP Poised promoter, PromBiv bivalent regulatory states, ReprPC repressed polycomb states and Quies a quiescent state. **c** Mean Pearson correlation of expression levels for gene pairs whose promoters were involved in PmPmI. The red dash line represents the observed mean Pearson correlation value of the significant PmPmI gene pairs and the histograms represent the null distributions of mean Pearson correlation values generated by repeated resampling of a set of distance-matched control gene pairs either from the genes whose promoters were involved in the SMR analysis (green) or from all genes (black). **d** Proportion of Pm-PAI genes in five gene activity groups with the first group being the inactive group (TPM < 0.1) together with four quartiles defined based on the expression levels of all genes in the GTEx blood samples. Data are presented as mean $+/-$ SD. The error bar represents the SD estimated from the 1000 control sets.

DNAm sites tested in the PAI analysis (1.14-fold on average, empirical $P_{enrichment} < 0.001$) or a random set of genomic sites (3.81-fold on average, empirical $P_{enrichment} < 0.001$; Supplementary Fig. 7a).

It has been shown in prior work that allelic imbalance in DNAm plays an important role in transcriptional regulation[30]. We thus tested whether the top associated mQTLs of the DNAm sites that showed significant PAIs were enriched for variants associated with allele-specific DNAm identified from Onuchic et al.[30]. There were 385 PAI mQTLs overlapping with variants associated with allele-specific DNAm, and the overlap was significantly larger than that of the same number of mQTLs randomly sampled from the mQTLs used in the PAI analysis (1.44-fold, empirical $P_{enrichment} < 0.001$; Supplementary Fig. 7b).

**Relevance of the predicted PAIs with gene expression.** We then turned to test whether pairwise genes with significant PmPmI were enriched for co-expression. We used gene expression data (measured by Transcript Per Kilobase Million mapped reads or TPM) from the blood samples of the Genotype-Tissue Expression

(GTEx) project[31] and computed the Pearson correlation of expression levels across individuals between pairwise genes ($r_P$). To assess the statistical significance of the enrichment, we compared the observed mean Pearson correlation of all the significant PmPmI gene pairs ($m = 2236$) to a null distribution of mean Pearson correlation values, generated by resampling a set of distance-matched control gene pairs either from the genes whose promoters were involved in the SMR analysis or from all genes. The mean correlation for the significant PmPmI gene pairs ($\bar{r}_P$) was 0.367, significantly (empirical $P_{enrichment} < 0.001$) higher than that for the control gene pairs sampled either from the genes whose promoters were involved in SMR (mean $\bar{r}_P = 0.292$; Fig. 4c) or from all genes (mean $\bar{r}_P = 0.156$; Fig. 4c), suggesting that pairwise genes with PmPmI are more likely to be co-expressed.

We also tested whether genes whose promoters were involved in significant PAI (called Pm-PAI genes hereafter, Fig. 1) were expressed more actively than the same number of control genes randomly sampled either from the genes whose promoters were involved in SMR or from all genes. Similar to the analysis above, we used the gene expression data from the blood samples of the GTEx project and tested the enrichment of Pm-PAI genes in different expression level groups (Methods). In comparison to the control sets sampled from the genes whose promoters were involved in SMR, Pm-PAI genes were significantly overrepresented (empirical $P_{enrichment} < 0.001$) among the group of genes with the highest expression levels and significantly underrepresented (empirical $P_{enrichment} < 0.001$) among genes that were not actively expressed (median TPM < 0.1) (Fig. 4d). These results implicate the regulatory role of the PIDSs in transcription and their asymmetric effects on gene expression. The enrichment was much stronger if the control sets were sampled from all genes (Supplementary Fig. 8a). We also performed a similar enrichment analysis (testing against the control sets sampled from all genes) for the predicted target genes from the PCHi-C data from Jung et al.[25]. There was a significant enrichment of the PCHi-C target genes in the active gene groups, but the fold enrichment was slightly smaller than that of the Pm-PAI genes (Supplementary Fig. 8), suggesting that PAIs could be more functionally relevant than PCHi-C loops.

**Enrichment of eQTLs in the PIDS regions.** We have shown that the PIDSs are located in regions enriched with regulatory elements (e.g., enhancers) (Fig. 4b) and that the Pm-PAI genes tend to have higher expression levels (Fig. 4d). We next investigated if genomic regions near PIDS are enriched for genetic variants associated with expression levels of the Pm-PAI genes using data from an expression QTL (eQTL) study in blood[32]. There were 11,204 independent cis-eQTLs at $P_{eQTL} < 5 \times 10^{-8}$ for 9967 genes, among which 2019 were Pm-PAI genes (Methods). We mapped cis-eQTLs to a 10 Kb region centred around each PIDS (5 Kb on either side) and counted the number of cis-eQTLs associated with expression levels of the corresponding Pm-PAI gene for each PIDS. There were 548 independent eQTLs located in the PIDS regions of the Pm-PAI genes, significantly higher than (empirical $P_{enrichment} < 0.001$) the mean of a null distribution (mean = 415) generated by randomly resampling distance-matched pairs of DNAm sites used in the SMR analysis (Fig. 5a). These results again imply the regulatory role of the PIDSs in transcription through eQTLs and provide evidence supporting the functional role of the predicted PAIs.

There were examples where a cis-eQTL was located in a PIDS region predicted to interact with the promoters of multiple genes. For instance, a cis-eQTL was located in an enhancer predicted to interact with the promoters of three genes (i.e., *ABCB9*, *ARL6IP4*

and *MPHOSPH9*) (Supplementary Fig. 9), and the predicted interactions were consistent with the TADs identified by Hi-C from Rao et al.[21] (Supplementary Fig. 3b). Furthermore, the predicted interactions between promoters of *ARL6IP4* and *MPHOSPH9* are consistent with the chromatin contact loops identified by Hi-C in the GM12878 cells[21] (Supplementary Fig. 9). The eQTL association signals were highly consistent for the three genes, and the pattern was also consistent with the SNP association signals for schizophrenia (SCZ) and years of education (EY) as shown in our previous work[19], suggesting a plausible mechanism whereby the SNP effects on SCZ and EY are mediated by the expression levels of at least one of the three co-regulated genes through the interactions of the enhancer and three promoters (Supplementary Fig. 9).

We have shown previously that the functional association between a DNAm site and a gene nearby can be inferred by the pleiotropic association analysis using SMR & HEIDI considering the DNAm level of a CpG site as the exposure and gene expression level as the outcome[19]. We further tested if the PIDSs are enriched among the DNAm sites showing pleiotropic associations with the expression levels of the neighbouring Pm-PAI genes. We found that approximately 15% of the PIDSs were the gene-associated DNAm sites identified in our previous study[19], significantly higher (empirical $P_{enrichment} < 0.001$) than that computed from the distance-matched control probe pairs (1.3%) described above (Fig. 5b).

**Replication of the predicted PAIs across tissues.** To investigate the robustness of the predicted PAIs across tissues, we performed the PAI analysis using brain mQTL data from the Religious Orders Study and Memory and Aging Project (ROSMAP)[33] ($n = 468$). Of the 11,082 PAIs with $P_{SMR} < 1.76 \times 10^{-9}$ and $P_{HEIDI} > 0.01$ in blood and available in brain, 2940 (26.5%) showed significant PAIs in brain after Bonferroni correction for multiple testing ($P_{SMR} < 4.51 \times 10^{-6}$ and $P_{HEIDI} > 0.01$). If we use a less stringent threshold for replication, e.g., the nominal $P$-value of 0.05, 66.31% of PAIs predicted in blood were replicated in brain. Here, the replication rate is computed based on a $P$-value threshold, which is dependent of the sample size of the replication data. Alternatively, we can estimate the correlation of PAI effects (i.e., the effect of the exposure DNAm site on the outcome DNAm site of a predicted PAI) between brain and blood using the $r_b$ method[34]. This method does not rely on a $P$-value threshold and accounts for estimation errors in the estimated effects, which is therefore less dependent of the replication sample size. The estimate of $r_b$ was 0.527 (SE = 0.0051) for 11,082 PAIs between brain and blood, suggesting a relatively strong overlap in PAI between brain and blood.

It is of note that among the 2940 blood PAIs replicated at $P_{SMR} < 4.51 \times 10^{-6}$ and $P_{HEIDI} > 0.01$ in brain, there were 268 PAIs for which the PAI effects in blood were in opposite directions to those in brain (Supplementary Data 1). For example, the estimated PAI effect between the *SORT1* and *SYPL2* loci was 0.49 in blood and –0.86 in brain. This tissue-specific effect is supported by the difference in gene expression correlation (correlation of expression levels between *SORT1* and *SYPL2* was −0.07 in whole blood and −0.37 in brain frontal cortex; $P_{difference} = 0.0018$ by Fisher transformation) and by the difference in the chromatin state of the promoter of *SYPL2* between brain and blood (bivalent promoter in blood and active promoter in brain; Supplementary Fig. 10). Taken together, while there are tissue-specific PAIs, a substantial proportion of the predicted PAIs in blood are consistent with those in brain.

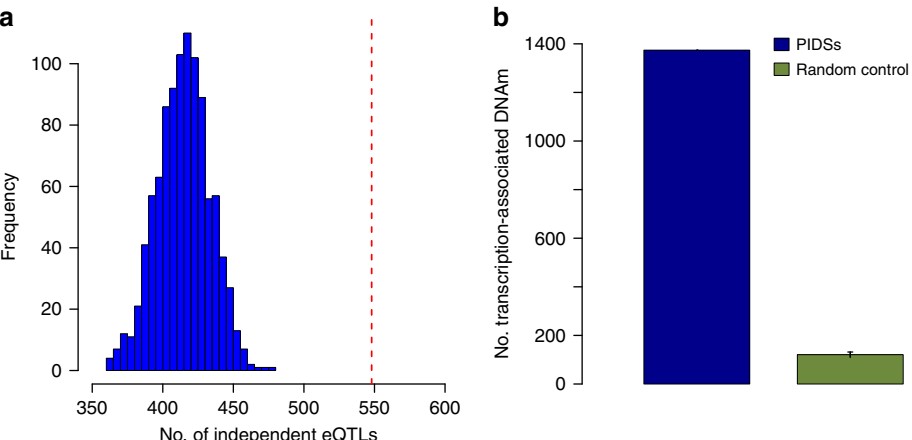

**Fig. 5 Enrichment of eQTLs or gene-associated DNAm sites in PIDS regions of the Pm-PAI genes. a** The number of independent cis-eQTLs ($P_{eQTL} < 5 \times 10^{-8}$) located in PIDS regions of the Pm-PAI genes. The red dash line represents the observed number and the blue histogram represents the distribution of 1000 control sets. **b** The number of transcription-associated DNAm sites located in PIDS regions of the Pm-PAI genes. The blue bar represents the observed number and the green bar represents the mean of 1000 control sets. Data are presented as mean +/− SD. The error bar represents the SD estimated from the 1000 control sets.

**Putative target genes of the disease-associated PIDSs**. We have shown above the potential functional roles of the predicted PAIs in transcriptional regulation. We then turned to ask how the predicted PAIs can be used to infer the genetic and epigenetic regulatory mechanisms at the GWAS loci for complex traits and diseases. We have previously reported 1203 pleiotropic associations between 1045 DNAm sites and 15 complex traits and diseases by an integrative analysis of mQTL and GWAS data using the SMR & HEIDI approach[19]. Of the 1045 trait-associated DNAm sites, 601 (57.5%) sites were involved in the predicted PAIs related to 299 Pm-PAI genes (Supplementary Data 2). We first tested the enrichment of the Pm-PAI genes of the trait-associated PIDSs using FUMA[35]. For the 15 complex traits analysed in Wu et al.[19], our FUMA analyses identified enrichment in multiple GO and KEGG pathways relevant to the corresponding phenotypes such as the inflammatory response pathway for Crohn's disease (CD) and steroid metabolic process for body mass index (BMI) (Supplementary Data 3), demonstrating the regulatory role of the trait-associated PIDSs in biological processes and tissues relevant to the trait or disease.

There were a number of examples where the predicted PAIs provided important insights to the functional genes underlying the GWAS loci and the underlying mechanisms by which the DNA variants affect the trait through genetic regulation of gene expression. One notable example was a PIDS (cg00271210) in an enhancer region predicted to interact in 3D space with the promoter regions of two genes (i.e., *RNASET2* and *RPS6KA2*), the expression levels of both of which were associated with ulcerative colitis (UC) and CD as reported in our previous study[19] (Fig. 6). The SNP association signals were consistent across CD GWAS, eQTL and mQTL studies, suggesting that the genetic effect on CD is likely to be mediated through epigenetic regulation of gene expression. Our predicted PAIs further implicated a plausible mechanism whereby the expression levels of *RNASET2* and *RPS6KA2* are co-regulated through the interactions of their promoters with a shared enhancer (Fig. 6), although only 41.5% of the predicted PAIs in this region overlapped with the TADs identified by Hi-C from the Rao et al. study[21] (Supplementary Fig. 3a) as mentioned above. According to the functional annotation data derived from the REMC samples, it appears that this shared enhancer is highly tissue-specific and presents only in B-cell and digestive system that are closely relevant to CD (Fig. 6). The over-expression of *RNASET2* in spleen (Supplementary

Fig. 11) is additional evidence supporting the functional relevance of this gene to CD. Another example is the *ATG16L1* locus (Supplementary Fig. 12). We have shown previously that five DNAm sites are in pleiotropic associations with CD and the expression level of *ATG16L1*[19]. Of these five DNAm sites, three were in an enhancer region and predicted to interact in 3D space with two DNAm sites in the promoter region of *ATG16L1* (Supplementary Fig. 12), suggesting a plausible mechanism that the genetic effect on CD at this locus is mediated by genetic and epigenetic regulation of the expression level of *ATG16L1* through promoter–enhancer interactions.

## Discussion

We have presented an analytical approach on the basis of the recently developed SMR & HEIDI method to predict promoter-anchored chromatin interactions using mQTL summary data. The proposed approach uses DNAm level of a CpG site in the promoter region of a gene as a bait to detect its pleiotropic associations with DNAm levels of the other CpG sites (Fig. 1) within 2 Mb distance of the focal promoter in either direction. In contrast to experimental assays, such as Hi-C and PCHi-C, our approach is cost-effective (because of the reuse of data available from experiments not originally designed for this purpose) and scalable to large sample sizes. Our method utilises a genetic model to perform a Mendelian randomisation analysis so that the detected associations are not confounded by non-genetic factors, which is also distinct from the methods that predict chromatin interactions from the correlations of chromatin accessibility measures[14,16]. The use of a genetic model to detect PAIs also facilitated the integration of the predicted PAIs with GWAS data (see Supplementary Note 1 for more discussion).

Using mQTL summary-level data from human peripheral blood ($n = 1980$), we predicted 34,797 PAIs for the promoter regions of 4617 genes. We showed that the predicted PAIs were enriched in TADs detected by published Hi-C and PCHi-C assays and that the PIDS regions were enriched with eQTLs of target genes. We also showed that the PIDSs were enriched in enhancers and that the Pm-PAI genes tended to be more actively expressed than matched control genes. These results demonstrate the functional relevance of the predicted PAIs to transcriptional regulation and the feasibility of using data from genetic studies of chromatin status to infer three-dimensional chromatin

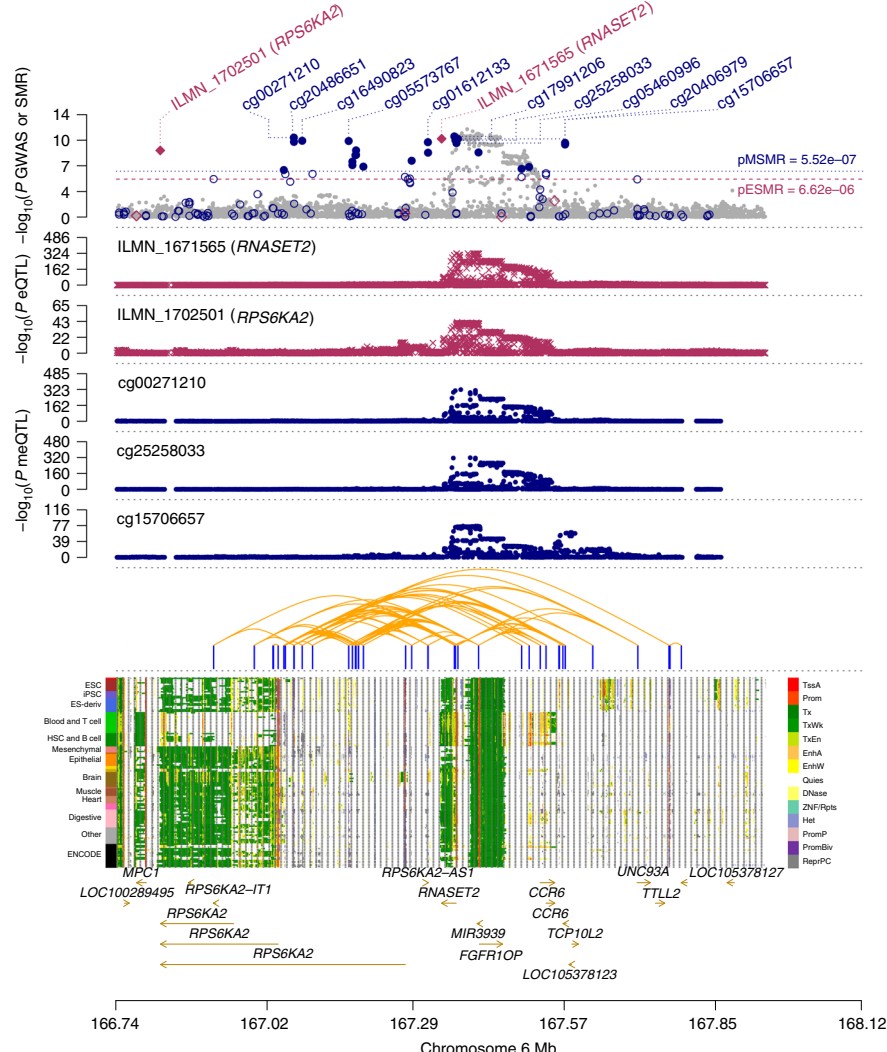

**Fig. 6 Prioritising genes and functional regions at the *RPS6KA2* locus for Crohn's disease (CD).** The top plot shows −log₁₀(*P*-values) of SNPs from the GWAS meta-analysis (grey dots) for CD[49]. Red diamonds and blue circles represent −log₁₀(*P*-values) from SMR tests for associations of gene expression and DNAm probes with CD, respectively. Solid diamonds and circles are the probes not rejected by the HEIDI test ($P_{HEIDI} > 0.01$). The second and third plots show −log₁₀(*P*-values) of SNP associations for the expression levels of probe ILMN_1671565 (tagging *RNASET2*) and ILMN_1702501 (tagging *RPS6KA2*), respectively, from the CAGE data. The fourth, fifth and sixth plots shows −log₁₀(*P*-values) of SNP associations for the DNAm levels of probes cg00271210, cg25258033 and cg15706657, respectively, from the mQTL meta-analysis. The panel on the bottom shows 14 chromatin state annotations (indicated by colours) inferred from 127 REMC samples (one sample per row) with the predicted PAIs annotated by orange curved lines on the top (see Supplementary Fig. 3a for the overlap of the predicted PAIs with Hi-C data).

interactions. The proposed approach is applicable to data from genetic studies of other chromatin features such as histone modification (i.e., hQTL)[36] or chromatin accessibility (caQTL)[26].

There are several reasons why the overlaps between the predicted PAIs and Hi-C loops were limited. First, Hi-C loops were detected with errors. We observed that the concordances between different Hi-C data sets were very limited (Supplementary Fig. 13), consistent with the conclusion from Forcato et al.[37] that the reproducibility of Hi-C loops is low at all resolutions. Second, most (65%) of our predicted PAIs are interactions between DNAm sites within 50 Kb (Supplementary Fig. 2b), which are often not well captured by the 3C-based methods due to its low resolution[17]. Third, the chromatin interactions are cell type specific[5] so that differences between the Hi-C loops identified in cell lines and our PAIs identified in whole blood are expected. For the PAIs that were between DNAm sites not located in TADs or Hi-C loops, we have shown specific examples that these predicted PAIs are likely to be functionally interacted (Fig. 2d and

Supplementary Fig. 3), suggesting that these PAIs are likely to be interactions yet to be identified by experimental assays. On the other hand, compared to the loops identified based on 3C-based methods, our predicted PAIs are more likely to be functional interactions due to the use of genetic and regulatory epigenomic data, as evidenced by the observation that our predicted Pm-PAI genes showed stronger enrichment in active gene groups compared to the predicted target genes from the PCHi-C data (Supplementary Fig. 8).

There are some limitations of this study. First, chromatin interactions are likely to be tissue- and temporal-specific, whereas our PAI analyses were limited to mQTL data from blood and brain owing to data availability and thus were unable to detect PAIs in specific tissues or at different developmental stages. Second, although the sample size of our blood mQTL summary data is large (n = ~2000), the PAI analysis could be under-powered if the proportion of variance in exposure or outcome explained by the top associated cis-mQTL is small. Third, the

predicted PAIs are relatively sparse as illustrated in Fig. 2d because of the sparsity of the DNAm array used, the underlying hypothesis of the SMR method, and the stringent statistical significance level used to claim significant PAIs (see Supplementary Note 2 for more discussion). The sparsity of the predicted PAIs can be reduced by the use of mQTL data from large samples based on whole-genome bisulfite sequencing or accurate DNAm imputation in the future. Fourth, the functional annotation data derived from the REMC samples could potentially include noise due to the small sample sizes, leading to uncertainty in defining the bait promoter regions. Fifth, if the DNAm levels of two CpG sites are affected by two sets of causal variants in strong linkage disequilibrium (LD), these two DNAm sites will appear to be associated in the SMR analysis and the power of the HEIDI test to reject such an SMR association will be limited because of the high LD[18,19]. However, this phenomenon is likely to be rare given that most of the promoter-anchored DNAm sites were predicted to interact with multiple DNAm sites, which are very unlikely to be all caused by distinct sets of causal variants in high LD. Sixth, the predicted PAIs including those falling in chromatin loops and TAD regions were not necessarily functional interactions and need to be validated by functional assays in the future. Despite these limitations, our study provides a novel computational paradigm to predict PAIs from genetic effects on epigenetic markers with high resolution. Integrating of the predicted PAIs with GWAS, gene expression and functional annotation data provides novel insights into the regulatory mechanisms underlying GWAS loci for complex traits. The computational framework is general and applicable to other types of chromatin and histone modification data, to further decipher the functional organisation of the genome.

## Methods

**Predicting PAIs from mQTL data by the SMR and HEIDI analyses.** This study is approved by the University of Queensland Human Research Ethics Committee (approval number: 2011001173). We used summary-level mQTL data to test whether DNAm levels of two CpG sites are associated because of a set of shared causal variants. Mendelian Randomisation (MR) is an approach developed to test for the causal effect of an exposure and an outcome using a genetic variant as the instrumental variable[38,39]. Summary-data–based Mendelian Randomisation (SMR) is a variant of MR, originally designed to test for association between the expression level of a gene and a complex trait using summary-level data from GWAS and eQTL studies[18] and subsequently applied to test for associations between DNAm and gene expression and between DNAm and complex traits[19]. Here, we applied the SMR analysis to detect associations between DNAm sites. Let $x$ be an exposure DNAm, $y$ be an outcome DNAm and $z$ be an instrument SNP associated with exposure DNAm (e.g., $P_{mQTL} < 5 \times 10^{-8}$). The SMR estimate of the effect of exposure DNAm on the outcome DNAm (i.e., $\hat{b}_{xy}$) is the ratio of the estimated effect of instrument on exposure ($\hat{b}_{zx}$) and that on outcome ($\hat{b}_{zy}$), $\hat{b}_{xy} = \hat{b}_{zy}/\hat{b}_{zx}$, where $\hat{b}_{zx}$ and $\hat{b}_{zy}$ are available from the summary-level mQTL data. We specified the DNAm level of a probe within the promoter region of a gene as the exposure and tested its associations with the DNAm levels of other probes (outcomes) within 2 Mb of the exposure probe (Fig. 1 and Supplementary Fig. 1). We excluded the DNAm pairs with a promoter region from the analysis because the chromatin interactions identified from Hi-C are often between a promoter region and nearby regions (i.e., the interactions within a promoter region are not studied) and because it helps reduce the computational and multiple testing burdens. For a pair of probes in two different promoter regions, the one with higher variance explained by its top associated cis-mQTL was used as the exposure and the other one was used as the outcome. The SMR test could possibly be due to linkage (i.e., distinct sets of causal variants in LD, one set affecting the exposure and the other set affecting the outcome), which is less of biological interest in comparison with pleiotropy (i.e., the same set of causal variants affecting both the exposure and the outcome). We then applied the HEIDI (heterogeneity in dependent instruments) test to distinguish pleiotropy from linkage. In brief, the HEIDI test was developed to test against the null hypothesis that the two DNAm sites are affected by the same set of causal variants. This is equivalent to testing whether there is a difference between the $\hat{b}_{xy}$ estimated from any mQTL $i$ ($\hat{b}_{xy(i)}$) and that estimated from the top associated mQTL ($\hat{b}_{xy(top)}$). If we define the difference in estimate between $\hat{b}_{xy}$ at mQTL $i$ and that at top associated mQTL as $\hat{d}_i = \hat{b}_{xy(i)} - \hat{b}_{xy(top)}$, then for multiple mQTLs (i.e., top 20 associated mQTLs after

pruning out SNPs in very strong LD), we have $\hat{\mathbf{d}} \sim MVN(\mathbf{d}, \mathbf{V})$, where $\hat{\mathbf{d}} = \{\hat{d}_1, \cdots, \hat{d}_{20}\}$ and $\mathbf{V}$ is the covariance matrix that can be estimated using summary-level mQTL data and LD information from a reference panel[18] (we used the 1KGP-imputed Health and Retirement Study data as the LD reference in this study). Therefore, we can test the evidence for heterogeneity through evaluating whether $\mathbf{d} = 0$ using an approximate multivariate approach[40]. We rejected the SMR associations with $P_{HEIDI} < 0.01$. All these analyses have been implemented in the SMR software tool (http://cnsgenomics.com/software/smr). As the mQTL data for the exposure and the outcome were obtained from the same sample, we investigated whether the SMR and HEIDI test-statistics were biased by the sample overlap. To this end, we computed the phenotypic correlation between each pair of exposure and outcome probes, as well as the variance explained by the top associated cis-mQTL of each exposure probe, and performed the simulation based on these observed distributions (Supplementary Note 3). The simulation results showed that $P$-values from both SMR and HEIDI tests were evenly distributed under the null model without inflation or deflation (Supplementary Fig. 14). We have made all the PAIs analysis scripts publicly available at https://github.com/wuyangf7/PAI.

**Data used for the PAI analysis and quality controls.** The peripheral blood mQTL summary data were from the Brisbane Systems Genetics Study (BSGS)[41] ($n = 614$) and Lothian Birth Cohorts (LBC) of 1921 and 1936[42] ($n = 1366$). We performed a meta-analysis of the two cohorts and identified 90,749 DNAm probes with at least a cis-mQTL at $P_{mQTL} < 5 \times 10^{-8}$ (excluding the probes in the major histocompatibility complex (MHC) region because of the complexity of this region), of which 28,732 DNAm probes were in the promoter regions defined by the annotation data derived from 23 REMC blood samples (T-cell, B-cell and Hematopoietic stem cells). The prefrontal cortex mQTL summary data were from the Religious Orders Study and Memory and Aging Project (ROSMAP)[33] ($n = 468$), comprising 419,253 probes and approximate 6.5 million genetic variants. In the ROSMAP data, there were 67,995 DNAm probes with at least a cis-mQTL at $P_{mQTL} < 5 \times 10^{-8}$ (not including the probes in the MHC region), of which 22,285 DNAm probes were in the promoter regions defined by the annotation data derived from 10 REMC brain samples. The mQTL effects were all in standard deviation (SD) units of DNAm levels. In the SMR analysis, the promoter DNAm site was used as the exposure and each of the other DNAm sites in a 2 Mb window was used as the outcome (Fig. 1). Note that we limited the analysis to a 2 Mb window because chromatin interactions between genomic sites >2 Mb apart are rare[21], because summary data from epigenetic QTL studies are often only available for genetic variants in cis-regions, and because it reduces the computational and multiple testing burdens. For the exposure probes, we included in the SMR analysis only the DNAm sites with at least one cis-mQTL (SNPs within 2 Mb of the CpG site associated with variation in DNAm level) at $P_{mQTL} < 5 \times 10^{-8}$. We used such a stringent significance level because a basic assumption of Mendelian randomisation is that the SNP instrument needs to be strongly associated with the exposure[38,39]. For all the DNAm probes, enhanced annotation data from Price et al.[43] (https://www.ncbi.nlm.nih.gov/geo/query/acc.cgi?acc=GPL16304) were used to annotate the closest gene of each DNAm probe.

We included in the analysis 15 complex traits (including disease) as analysed in Wu et al.[19]. They are height[44], body mass index (BMI)[45], waist-hip-ratio adjusted by BMI (WHRadjBMI)[46], high-density lipoprotein (HDL)[47], low-density lipoprotein (LDL)[47], thyroglobulin (TG)[47], educational years (EY)[48], rheumatoid arthritis (RA)[49], schizophrenia (SCZ)[50], coronary artery disease (CAD)[51], type 2 diabetes (T2D)[52], Crohn's disease (CD)[53], ulcerative colitis (UC)[53], Alzheimer's disease (AD)[54] and inflammatory bowel disease (IBD)[53]. The GWAS summary data were from the large GWAS meta-analyses (predominantly in samples of European ancestry) with sample sizes of up to 339,224. The number of SNPs varied from 2.5 to 9.4 million across traits.

**Annotations of the chromatin state.** The epigenomic annotation data used in this study were from the Roadmap Epigenomics Mapping Consortium (REMC), publicly available at http://compbio.mit.edu/roadmap/. We used these data to annotate the functional relevance of the DNAm sites and their cell type or tissue specificity. The chromatin state annotations from the Roadmap Epigenomics Project[13] were predicted by ChromHMM[12] based on the imputed data of 12 histone modification marks. It contains 25 functional categories for 127 epigenomes in a wide range of primary tissue and cell types. The 25 chromatin states were further combined into 14 main functional annotations (as shown in Fig. 4b and Wu et al.[19]).

**Enrichment of the predicted PAIs in chromatin contacts.** To test the enrichment of our predicted PAIs in chromatin contacts detected by Hi-C, PCHi-C or ChIA-PET, we used chromatin contact loops and topological-associated domains (TADs) data from the Rao et al.[21] study called in the GM12812 cells and the Dixon et al.[23] study in embryonic stem cells, PCHi-C interaction data generated from human primary hematopoietic cells[5], and the POLR2A ChIA-PET chromatin loops from the ENCODE project[24] (Supplementary Table 1). To assess the statistical significance of the enrichment, we generated a null distribution by randomly sampling 1000 sets of control probe pairs (with the same number of control probe pairs as that of the predicted PAIs in each set) from the distance-matched probe pairs

tested in the SMR analysis. We mapped both the predicted PAIs and the control probe pairs to the TAD regions or chromatin contact loops detected by previous experimental assays and quantified the number of overlapping pairs. We estimated the fold enrichment by the ratio of the overlapping number for the predicted PAIs to the mean of the null distribution and computed the empirical $P$-value by comparing the overlapping number for the predicted PAIs with the null distribution.

We used the chromatin interaction data generated by Jung et al.[25] in GM12878 cell lines as a validation set to evaluate the performance of different interaction prediction methods. We computed the enrichment of the predicted interactions by different methods in the significant PCHi-C loops defined based on a range of PCHi-C $P$-value thresholds with a $2 \times 2$ contingency table and used the Fisher's exact test to assess the statistical significance of the enrichment.

**Enrichment of the PIDSs in functional annotations**. To conduct an enrichment test of the promoter-interacting DNAm sites (PIDSs) in different functional annotation categories, we first extracted chromatin state data of 23 blood samples from the REMC samples. We then mapped the PIDSs to 14 main functional categories based on the physical positions and counted the number of PIDSs in each functional category. Again, we generated a null distribution by randomly sampling the same number of control probes (with variance in DNAm level matched with the PIDSs) from all the probes tested in the PAI analysis and repeated the random sampling 1000 times. The fold enrichment was calculated by the ratio of the observed value to the mean of the null distribution, and an empirical $P$-value was computed by comparing the observed value with the null distribution.

**Enrichment of the predicted PAIs in protein-DNA interactions**. We used the chromatin immuno-precipitation sequencing (ChIP-Seq) data from GM12878 for four DNA-binding proteins (i.e., CTCF, Rad21, ZNF143, YY1) from the ENCODE[24] project to test whether our predicted PAIs are enriched in the binding regions of proteins known to be involved in 3D organisation of the genome. There were 21,787 unique DNAm sites involved in the predicted PAIs. We mapped the ChIP-Seq peaks of the four DNA-binding proteins to a 10 Kb region centred around each PAI DNAm site. To test if the number of overlaps was significantly larger than expected by chance, we generated a null distribution by mapping the ChIP-Seq peaks of each protein to either the same number of DNAm sites randomly sampled from those included in the PAI analysis or the same number of random genomic sites.

**Quantifying the expression levels of Pm-PAI genes**. To quantify the expression levels of genes whose promoters were involved in the predicted PAIs (Pm-PAI genes), we used gene expression data (measured by transcript per kilobase million mapped reads (TPM)) from blood samples of the Genotype-Tissue Expression (GTEx) project[31] (https://www.gtexportal.org/home/). We classified all the genes into two groups based on their expression levels in GTEx blood, i.e., active and inactive (TPM < 0.1). For the active genes, we further divided them into four quartiles based on their expression levels in GTEx blood, and counted the number of Pm-PAI genes in each group. To generate the null distribution, we randomly sampled the same number of control genes whose promoter DNAm sites were included in the SMR analysis, and repeated the random sampling 1000 times. We computed the number of Pm-PAI genes and control genes in each group and assessed the significance by comparing the number of Pm-PAI genes with the null distribution in each group. We further tested the enrichment of the Pm-PAI genes against a null distribution sampled from all genes.

**Enrichment of eQTLs and gene-associated DNAm in the PIDSs**. The eQTL enrichment analysis was conducted using all the independent cis-eQTLs ($m = 11,204$) from the CAGE[32] study. The independent cis-eQTLs were from SNP-probe associations ($P < 5 \times 10^{-8}$) after clumping analysis in PLINK[55] followed by a conditional and joint (COJO) analysis in GCTA[56]. We only retained the cis-eQTLs whose target genes had at least a PIDS and mapped the cis-eQTL to a 10 Kb region centred around each corresponding PIDS of a Pm-PAI gene. To assess the significance of the enrichment, we generated a null distribution by mapping the cis-eQTLs to the same number of control gene-DNAm pairs (strictly speaking, it is the bait DNAm probe in the promoter of a gene together with another non-promoter DNAm probe) randomly sampled (with 1,000 repeats) from those included in the PAI analysis with the distance between a control pair matched with that between a Pm-PAI gene and the corresponding PIDS. In addition, we have identified a set of DNAm sites that showed pleiotropic associations with gene expressions in a previous study[19]. We used the same approach as described above to test the significance of enrichment of the gene-associated DNAm sites in the PIDSs.

**Reporting summary**. Further information on research design is available in the Nature Research Reporting Summary linked to this article.

## Data availability

All the data sets used in this study are available in the public domain (Supplementary Table 1). The full summary statistics from the PAI analysis are publicly available at http://cnsgenomics.com/shiny/M2Mdb/.

## Code availability

All the analysis scripts used in this study are publicly available at https://github.com/wuyangf7/PAI.

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

## Acknowledgements

We thank Peter Visscher for helpful discussion. This research was supported by the Australian Research Council (DP160101343, DP160101056 and FT180100186), the Australian National Health and Medical Research Council (1107258, 1083656, 1078901 and 1113400), and the Sylvia & Charles Viertel Charitable Foundation. The Lothian Birth Cohorts (LBC) are supported by Age UK (Disconnected Mind programme). Methylation typing was supported by Centre for Cognitive Ageing and Cognitive Epidemiology (Pilot Fund award), Age UK, The Wellcome Trust Institutional Strategic Support Fund, The University of Edinburgh, and The University of Queensland. The LBC resource is prepared in the Centre for Cognitive Ageing and Cognitive Epidemiology, which is supported by the Medical Research Council and Biotechnology and Biological Sciences Research Council (MR/K026992/1), and which supports I.J.D. This study makes use of data from dbGaP (accessions: phs000428.v1.p1 and phs000424.v1.p1) and EGA (accession: EGAS00001000108). A full list of acknowledgements to these data sets can be found in the Supplementary Note **4**.

## Author contributions

J.Y. conceived and supervised the study. Y.W., T.Q. and J.Y. designed the experiment. Y.W. and T.Q. performed simulations and statistical analyses under the assistance or guidance from J.Y., J.Z., H.W., F.Z. and Z.Z. I.J.D., N.R.W. and A.F.M. contributed the blood DNA methylation data. J.E.P.C. provided critical advice that significantly improved the interpretation of the results. N.R.W. and J.Y. contributed funding and resources. Y.W., T.Q., J.Z. and J.Y. wrote the manuscript with the participation of all authors.

## Competing interests

The authors declare no competing interests.
