## [Peer Review File · Nature Communications]

Reviewers' Comments:

Reviewer #1:

Remarks to the Author:

Summary:

The authors propose a novel approach to predict promoter-anchored chromatin interactions using DNA methylation QTL summary data. Their approach relies on previously published methods called SMR and HEIDI that implement Mendelian randomization to remove confounding factors and account for linkage disequilibrium (LD) to fine map SNPs belonging to the same LD blocks. The article is well written and is easy to read. The contribution of this article is significant, since the approach is novel and bridge two different fields: 3D chromatin and statistical genetics. It would help to target promoter-anchored chromatin interactions that are involved in GWAS, therefore allowing a better interpretation of the SNP effect on disease.

Major revision:

1/ The authors must compare their prediction method with state-of-the-art methods for predicting long-range interactions, when possible. For instance, they can compute correlations between different DNA methylation probes (without accounting for genotype information) and show that their Mendelian randomization improves the results. A similar approach would be to compare with correlations between other kinds of chromatin data (histone mark ChIP-seq, protein binding ChIP-seq, DNase-seq, ...), or expression data (CAGE-seq from Fantom project or other expression seq data that map gene expression as well as enhancer expression with strand-specific data) from cell lines. Moreover, the author should compare their approaches using predictive models that predicts promoter-enhancer interactions using epigenomic data such as in Bing He et al. (Global view of enhancer-promoter interactome in human cells, PNAS May 27, 2014 111 (21) E2191-E2199), or any other modeling approach.

2/ Predictions based on DNA methylation seems to be quite sparse when compared to Hi-C data (Figure 2d) as discussed by the authors in the Discussion. The authors should explain if the sparsity of these predictions are due to the low density of probes along the genome, or are due to any other problem related to their method. They should illustrate more this problem in the result section. In this line, the authors should explain if their predictions are not biased toward certain regions of the genome due to the probe density or technical artefacts and illustrate with results.

3/ The authors should explain and illustrate with figures if their predictions are more related to general features of Hi-C data (for instance TADs or compartments) or more specific features such as loops identified from Hi-C data or ChIA-PET.

4/ The authors must add more explanations on SMR and HEIDI in the article, since I had to read carefully their previous article (Integration of summary data from GWAS and eQTL studies predicts complex trait gene targets, Nature Genetics, 2016) to understand how the novel method works.

5/ The authors should add some ROC and PR curves to evaluate the accuracy of PAI predictions. That would help also to compare with other methods (see comment 1/)

Reviewer #2:

Remarks to the Author:

The authors presented the prediction of promoter-anchored chromatin interactions using DNA methylation data. They proposed an analytical approach to convert 1D epigenetic information to 3D chromatin structure. Overall, this is a well-designed study. This study provided a novel insight

for studying chromatin 3D organization. Below are a few concerns:

1. It would be more user-friendly to package all scripts required to reproduce the work as supplemental materials and/or upload to GitHub.
2. In Page 4 Line 99, the authors limited the association analysis within "2 Mb" of the gene. Are there any statistical reasons for using "2 Mb" rather than other values?
3. In Page 5 Line 126, ChIA-PET could be an alternative method for mapping the chromatin 3D interaction. It would be better to show the overlap between ChIA-PET data (such as Pol2) and the PAIs of this study.
4. In Page 5 Line 126, the percentage of PAIs located in TADs of Hi-C data was not so high even though the statistical significance was shown. How to explain those PAIs that were not located in TADs? Are they false positives?
5. In the section "Enrichment of the predicted PAIs in functional annotations" (Page 6), the analyses were confusing. For those significantly-enriched regions (e.g., repressed Polycomb regions and high DNase sensitivity sites) that the authors claimed, the fold-enrichment value seems small (most were less than 2.0). In addition, it did not make sense that the PIDSs were underrepresented surrounding transcription start sites but significantly-enriched in the bivalent promoters.
6. In the section "Relevance of the predicted PAIs with gene expression" (Page 6-7), the authors analyzed the association between PmPmI and gene co-expression. However, in Figure 3C, the authors only included the mean value (red line) for PmPmI gene pairs while including the histogram for the "control". It is confusing why not including both distributions? In addition, the authors should point out which statistical test was used when they claimed the pairwise genes with PmPmI were more likely to be co-expressed.

Reviewer #3:

Remarks to the Author:

Review of "Promoter-anchored chromatin interactions predicted from genetic analysis of epigenomic data," by Yang Wu et al. This paper presents a statistical method to predict promoter associated interactions using covariation of DNA methylation and mQTL data across individuals. The approach is based on Mendelian randomization and HEIDI and used to analyze mQTL data from a meta-analysis of studies on 1,980 individuals to predict the interactions between promoters and genomic regions within 4Mbp of the promoters. Our major concern is that the evidence in support of the predicted interactions is quite weak. Although anecdotal evidence is presented from a few gene loci, these are not well annotated and hand-picked examples where the effect of covarying methylation should be more obviously functionally significant. There is no direct experiment validation of novel interactions, and no direct comparison to previously published interaction datasets, which could be done by comparing the full prediction sets to promoter capture hi-C or hi-chip. Only one hi-c interaction is shown in a supplemental figure. Beyond that there are no hard examples shown that differential methylation is recovering known demonstrated E-P interactions. While the predictions do fall within TADs more frequently by chance, this is not a direct assessment of functional interaction. The enrichment for differentially expressed genes in fig3d looks marginally significant at best.

Minor:

1. All plots are poorly annotated and difficult to follow. In Fig 1a, where do probe 1 and probe 2 map to on Fig 1b? Also in Fig 1b, are so many acronyms really necessary: PAI, PIDS, PmPmI, PmPAI.
2. It should be noted that earlier papers have observed co-varying accessibility (ATAC-seq) and noted that these co-varying peaks could imply direct physical interactions (Kumasaka et al Nature Genetics 2016, Gate et al Nature Genetics 2018, ref 16 is low resolution but these two and

Kumasaka et al Nature Genetics 2019 are high resolution) in addition to the Kumasaka et al Nature Genetics 2019 paper.

REVIEWERS' COMMENTS

We thank the three reviewers for their constructive comments, which have helped us improve the manuscript substantially. We have responded to all the reviewers' comments point-by-point below (in blue) and have highlighted all the relevant changes (in yellow) in the revised manuscript.

Reviewer #1:

Summary:

The authors propose a novel approach to predict promoter-anchored chromatin interactions using DNA methylation QTL summary data. Their approach relies on previously published methods called SMR and HEIDI that implement Mendelian randomization to remove confounding factors and account for linkage disequilibrium (LD) to fine map SNPs belonging to the same LD blocks. The article is well written and is easy to read. The contribution of this article is significant, since the approach is novel and bridge two different fields: 3D chromatin and statistical genetics. It would help to target promoter-anchored chromatin interactions that are involved in GWAS, therefore allowing a better interpretation of the SNP effect on disease.

Re: We thank the reviewer for the positive remarks.

Major revision:

1. The authors must compare their prediction method with state-of-the-art methods for predicting long-range interactions, when possible. For instance, they can compute correlations between different DNA methylation probes (without accounting for genotype information) and show that their Mendelian randomization improves the results. A similar approach would be to compare with correlations between other kinds of chromatin data (histone mark ChIP-seq, protein binding ChIP-seq, DNase-seq, ...), or expression data (CAGE-seq from Fantom project or other expression seq data that map gene expression as well as enhancer expression with strand-specific data) from cell lines. Moreover, the author should compare their approaches using predictive models that predicts promoter-enhancer interactions using epigenomic data such as in Bing He et al. (Global view of enhancer-promoter interactome in human cells, PNAS May 27, 2014 111 (21) E2191-E2199), or any other modeling approach.

Re: We have compared our SMR & HEIDI method with two state-of-the-art methods in the revised manuscript (lines 156-184). We first compared our method with a correlation-based method (i.e., a method that uses correlations of epigenomic marks to predict interactions; Ernst *et al.*, 2011, *Nature*) using two different types of epigenomic data, i.e., DNA methylation (DNAm) and chromatin accessibility measured by Assay for Transposase-Accessible Chromatin using sequencing (ATAC-seq). To evaluate the performance of the methods, we used a recently released chromatin interaction data (PChi-C loops) generated by Jung *et al.* (2019, *Nature Genetics*) in GM12878 cell lines for validation, and quantified the enrichment of the predicted interactions in PChi-C loops defined based on a range of PChi-C *P* value thresholds. We chose the PChi-C data from Jung *et al.* because the *P* values of all the tested loops are available and because compared to other Hi-C data sets, chromatin interactions identified in GM12878 cell

lines may be more relevant to the predicted PAIs in whole blood. We computed the fold enrichment of our predicted PAIs in the PCHi-C loops by a 2×2 contingency table and used the Fisher's exact test to assess the statistical significance of the enrichment. The results showed that our predicted PAIs using either DNAm or chromatin accessibility data were highly enriched in the PCHi-C loops and that the fold enrichment increased with the increase of the significance level used to claim the PCHi-C loops (**Fig. 3c**), consistent with the observation from previous work that Hi-C loops with lower P values are more reproducible between biological replicates (Jin *et al.*, 2013, *Nature*). Our SMR & HEIDI method outperformed the correlation-based method using either DNAm or chromatin accessibility data, as evidenced by the larger fold enrichment of our method compared to the correlation-based method at all the PCHi-C significance levels (**Fig. 3c**).

As pointed out by the reviewer, there are other predictive models such as the method developed by He *et al.* (2014, *PNAS*) that uses multiple genomic features to predict specific chromatin interactions (i.e., enhancer-promoter interactions). Considering that the He *et al.*'s method and our method required very different types of data and that our predicted PAIs are not restricted to enhancer-promoter interactions, we did not compare the two methods here. Instead, we compared our SMR & HEIDI method with the pairwise hierarchical model (PHM) that also uses genetic data of regulatory elements (Kumasaka *et al.*, 2019, *Nature Genetics*). Applying our SMR & HEIDI method to the summary-level chromatin accessibility QTL (caQTL) data from Kumasaka *et al.*, of the 15,487 causal interactions identified by the PHM approach, 10,416 were tested in our SMR & HEIDI analysis; 98.4% were replicated at a nominal significance level ($P_{\text{SMR}} < 0.05$ and $P_{\text{HEIDI}} > 0.01$), and 36% were significant after multiple testing corrections ($P_{\text{SMR}} < 4.8 \times 10^{-6}$ ($0.05/10,416$) and $P_{\text{HEIDI}} > 0.01$). While the PHM method requires individual-level genotype and chromatin accessibility data and is less computationally efficient due to the use of Bayesian hierarchical model, our SMR & HEIDI method that requires only summary-level data is more flexible and can be potentially applied to all epigenetic QTL data. We have added these results in the revised manuscript (**Fig. 3c** and lines 156-184).

2. Predictions based on DNA methylation seems to be quite sparse when compared to Hi-C data (Figure 2d) as discussed by the authors in the Discussion. The authors should explain if the sparsity of these predictions are due to the low density of probes along the genome, or are due to any other problem related to their method. They should illustrate more this problem in the result section. In this line, the authors should explain if their predictions are not biased toward certain regions of the genome due to the probe density or technical artefacts and illustrate with results.

Re: In Figure 2d, we compared the predicted PAIs selected at a very stringent significance level (i.e., the experiment-wise significance level) to the chromatin interactions with correlation scores > 0.4 from Grubert *et al.* (2015, *Cell*), which does not provide a fair comparison of sparsity between PAIs and Hi-C loops in this region. We have clarified this in the revised manuscript (**Fig. 2**).

Nevertheless, we acknowledge that the predicted PAIs are relatively sparse because of the sparsity of the DNAm array used, the underlying hypothesis of the SMR method, and the

stringent statistical significance level used to claim significant PAIs (lines 409-412 and **Supplementary Note 1**). More specifically, first, although the Illumina 450K methylation array has a genome-wide coverage, the probes cover only a limited proportion of the regulatory elements. Second, SMR requires the exposure probe with at least an mQTL at $P_{mQTL} < 5e-8$, and we limited the exposure probes in promoter regions, resulting in only a small proportion of exposure DNAm probes ($m=28,732$, $\sim 6.5\%$) being included in the SMR & HEIDI analysis. Third, to control for false positives, we applied an experiment-wise SMR significance threshold (i.e., $P_{SMR} < 1.76e-9$) to correct for multiple testing and a stringent HEIDI threshold (i.e., $P_{HEIDI} < 0.01$) to reject SMR associations due to linkage. However, despite the relatively sparse distribution of the predicted PAIs across the genome, the number of predicted PAIs ($m=34,797$) is comparable to the loops identified by experimental assays such as Hi-C and PCHi-C. For example, there are only $\sim 10,000$ Hi-C loops identified from Rao *et al.* (2014, *Cell*) and $\sim 80,000$ PCHi-C loops identified from Jung *et al.* (2019, *Nature Genetics*).

In addition to the sparsity, the Illumina 450K DNAm probes are preferentially distributed towards certain genome regions (e.g., promoter; **Fig 4a**). Such an uneven distribution, however, would not bias the functional enrichment results of our predicted PAIs (e.g., those shown in **Figs. 4 and 5**) because the enrichments were tested against DNAm pairs randomly sampled from all the pairs tested in the SMR & HEIDI analysis rather than random genomic positions. We have commented on this in the revised manuscript (lines 200-204).

3. The authors should explain and illustrate with figures if their predictions are more related to general features of Hi-C data (for instance TADs or compartments) or more specific features such as loops identified from Hi-C data or ChIA-PET.

Re: Our predicted PAIs are more related to the general features of Hi-C data such as topologically associating domains (TADs) than the specific features, as suggested by the three observations below. First, $\sim 80\%$ of the predicted PAIs were located in the TADs identified from Dixon *et al.* (2012, *Nature*) in comparison to only 130 PAIs overlapped with the $\sim 10,000$ Hi-C loops identified from Rao *et al.* (2014, *Cell*). Second, the fold enrichment of the predicted PAIs in TADs (1.89-fold) was larger than that in specific Hi-C loops (1.49-fold) using the same Hi-C data from Rao *et al.* (**Fig. 2a**). Third, we performed an additional enrichment analysis of the predicted PAIs in the *POLR2A* ChIA-PET loops from the ENCODE and observed a significant but smaller enrichment of the predicted PAIs in ChIA-PET loops (1.44-fold, one-sided empirical $P < 0.001$, **Fig. 3b**) than that in TADs.

There are several reasons why the overlaps between the PAIs and Hi-C loops were limited. First, Hi-C loops were detected with errors. We observed that the concordances between different Hi-C data sets were very limited (**Fig. S11**), consistent with the conclusion from Forcato *et al.* (2017, *Nature Methods*) that the reproducibility of Hi-C loops is low at all resolutions. Second, most (65%) of our predicted PAIs are interactions between DNAm sites within 50 Kb (**Fig. S2b**), which are often not well captured by the 3C-based methods due to its low resolution (Kumasaka *et al.*, 2019, *Nature Genetics*). Third, Hi-C loops are cell type specific (Javierre *et al.*, 2016, *Cell*) so that differences between the Hi-C loops identified in cell lines and our PAIs identified in whole blood are expected. We have discussed this issue in the revised manuscript (lines 388-402).

4. The authors must add more explanations on SMR and HEIDI in the article, since I had to read carefully their previous article (Integration of summary data from GWAS and eQTL studies predicts complex trait gene targets, Nature Genetics, 2016) to understand how the novel method works.

Re: We have added more explanations on the SMR & HEIDI method in the Methods section (lines 439-443 and 453-462).

5. The authors should add some ROC and PR curves to evaluate the accuracy of PAI predictions. That would help also to compare with other methods (see comment 1).

Re: We agree with the reviewer that ROC curves are useful for method comparison. In our case, however, the quantification of the specificity and sensitivity is hindered by the lack of ground truth positive and negative controls of the chromatin interactions. There were several reasons why there is no ideal Hi-C data set that could be used to conduct the ROC analysis. First, the Hi-C loops were detected with errors. The concordances between different Hi-C data sets were very limited (**Fig. S11**). Second, most of the Hi-C maps were generated at > 5 Kb resolution, while our predicted PAIs were interactions between two DNAm sites at single base-pair resolution. Therefore, it is very likely that multiple PAIs with different significance levels could be mapped to the same Hi-C loop. In this case, it is difficult to evaluate specificity and sensitivity of the prediction. Third, Hi-C loops are cell-type specific so that there are expected differences between the Hi-C loops identified in cell lines and our PAIs identified in whole blood.

Instead, we have compared the interaction prediction methods by testing the enrichment of the predicted interactions in chromatin loops defined based on a range of Hi-C *P* value thresholds. As mentioned above (Remark 1), we used a recently published PChi-C data set in GM12878 cell lines from Jung *et al.* (2019, *Nature Genetics*) because the *P* values of all the tested loops are available and because compared to other Hi-C data sets, chromatin interactions identified in GM12878 cell lines may be more relevant to the predicted PAIs in whole blood. The result showed that our predicted PAIs were highly enriched in the Jung *et al.* PChi-C data and that the fold enrichment increased with the increase of the significance level used to claim the PChi-C loops (**Fig. 3c**). Moreover, our SMR & HEIDI method outperformed the correlation-based method using either DNAm or chromatin accessibility data and is more computationally efficient and flexible in comparison to another genetic data based prediction method (i.e., the PHM approach).

Reviewer #2

(Remarks to the Author):

The authors presented the prediction of promoter-anchored chromatin interactions using DNA methylation data. They proposed an analytical approach to convert 1D epigenetic information to 3D chromatin structure. Overall, this is a well-designed study. This study provided a novel insight for studying chromatin 3D organization. Below are a few concerns:

1. It would be more user-friendly to package all scripts required to reproduce the work as supplemental materials and/or upload to GitHub.

Re: We have made all the PAI analysis scripts publicly available at GitHub <https://github.com/wuyangf7/PAI> (lines 470-471).

2. In Page 4 Line 99, the authors limited the association analysis within “2 Mb” of the gene. Are there any statistical reasons for using “2 Mb” rather than other values?

Re: We limited the PAI analysis to DNAm pairs within a 2Mb window for the following reasons (see lines 99-102 in the revised manuscript). First, we knew from the Hi-C data that chromatin interactions between genomic sites separated by more than 2 Mb are very rare (Jin *et al.*, 2013, *Nature*). Second, summary data from epigenetic QTL studies are often only available for genetic variants in cis-regions. Third, the use of a 2Mb window reduces the computational and multiple testing burdens. In fact, our results showed that only ~0.7% of the predicted PAIs were between DNAm sites greater than 1 Mb apart (lines 122-123).

3. In Page 5 Line 126, ChIA-PET could be an alternative method for mapping the chromatin 3D interaction. It would be better to show the overlap between ChIA-PET data (such as Pol2) and the PAIs of this study.

Re: We thank the reviewer for this suggestion. We have performed the analysis to test whether our predicted PAIs were also enriched in chromatin interactions identified by ChIA-PET (lines 150-151). We used the *POLR2A* ChIA-PET data from the ENCODE project. There were ~2,300 PAIs overlapping with the ChIA-PET loops, and the number of overlaps was significantly higher than that of the same number of DNAm pairs randomly sampled from all the tested DNAm pairs with distances matched (1.44-fold, one-sided empirical *P*-value < 0.001, **Fig. 3b**).

4. In Page 5 Line 126, the percentage of PAIs located in TADs of Hi-C data was not so high even though the statistical significance was shown. How to explain those PAIs that were not located in TADs? Are they false positives?

Re: First, we would like to clarify that the detected TAD regions are not perfect because they are predicted by computational approaches with errors/uncertainty. For example, Dali *et al.* (2017, *Nucleic Acids Res*) concluded that the predicted TADs varied greatly among prediction tools and datasets in number, size, and other biological properties. Second, since we have applied an experiment-wise significance level to correct for multiple testing in the PAI analysis, the false positive rate is expected to be very low (a probability of 0.05 to observe one or more false positives in the whole study). This is also supported by the result that ~80% of the PAIs were between DNAm within TADs (lines 134-137). For the PAIs that were between DNAm not located in any TADs, we have shown specific examples that these predicted PAIs are likely to be functionally interacted (**Fig. 2d** and **Fig. S3**), suggesting that they are not false positives but likely to be interactions yet to be identified by experimental assays.

5. In the section “Enrichment of the predicted PAIs in functional annotations” (Page 6), the analyses were confusing. For those significantly-enriched regions (e.g., repressed Polycomb regions and high DNase sensitivity sites) that the authors claimed, the fold-enrichment value seems small (most were less than 2.0). In addition, it did not make sense that the PIDs were underrepresented surrounding transcription start sites but significantly-enriched in the bivalent promoters.

Re: The fold-enrichment was computed as the proportion of promoter-interacting DNAm sites (PIDs) in a functional category divided by the mean of a null distribution generated by resampling variance-matched control probes at random from all the outcome probes used in the SMR analysis. On one hand, the enrichment test is not biased by the fact that the Illumina 450K methylation array probes are preferentially distributed towards certain genomic regions because it tests against control probes sampled from those on the array rather than random genomic positions. On the other hand, however, this test is over conservative because the control probes are enriched in certain functional genomic regions (**Fig. S5a**) and can possibly contain some of the PIDs, which may explain the relatively small fold enrichments observed in this analysis. We have clarified this in the revised manuscript (lines 200-208).

In the PAI analysis, we excluded the DNAm pairs within a promoter region, which may explain why the PIDs were depleted in promoters. If we add the within-promoter DNAm pairs back in analysis, the predicted PIDs are significantly enriched in both promoters and bivalent promoters (**Fig. S5b**). We have commented on this in the revised manuscript (lines 206-208).

6. In the section “Relevance of the predicted PAIs with gene expression” (Page 6-7), the authors analysed the association between PmPmI and gene co-expression. However, in Figure 3C, the authors only included the mean value (red line) for PmPmI gene pairs while including the histogram for the “control”. It is confusing why not including both distributions? In addition, the authors should point out which statistical test was used when they claimed the pairwise genes with PmPmI were more likely to be co-expressed.

Re: The method used in this analysis is an empirical test that compares the observed mean Pearson correlation of all the PmPmI gene pairs to the distribution of a number of mean Pearson correlation values under the null. This null distribution was generated by randomly sampling a distance-matched control set from the gene pairs whose promoter were tested in the SMR analysis for 1,000 times. So, it is a comparison of the observed mean value with the null distribution of the mean values. We have clarified this in the main text (lines 217-223) and the legend of Figure 3c (now Figure 4c in the revised manuscript).

Reviewer #3

(Remarks to the Author):

Review of “Promoter-anchored chromatin interactions predicted from genetic analysis of epigenomic data,” by Yang Wu et al. This paper presents a statistical method to predict

promoter associated interactions using covariation of DNA methylation and mQTL data across individuals. The approach is based on Mendelian randomization and HEIDI and used to analyse mQTL data from a meta-analysis of studies on 1,980 individuals to predict the interactions between promoters and genomic regions within 4Mbp of the promoters.

Re: We thank the reviewer for the summary.

Our major concern is that the evidence in support of the predicted interactions is quite weak. Although anecdotal evidence is presented from a few gene loci, these are not well annotated and hand-picked examples where the effect of covarying methylation should be more obviously functionally significant. There is no direct experiment validation of novel interactions, and no direct comparison to previously published interaction datasets, which could be done by comparing the full prediction sets to promoter capture hi-C or hi-chip. Only one hi-c interaction is shown in a supplemental figure.

Re: To further validate our method, we have compared our predicted PAIs with the chromatin loops identified by additional experimental assays (i.e., promoter capture Hi-C (PChI-C) and Chromatin Interaction Analysis by Paired-End Tag Sequencing (ChIA-PET)), and compared our method with two other prediction approaches, i.e., the correlation-based method from Ernst *et al.* (2011, *Nature*) and the pairwise hierarchical model (PHM) method from Kumasaka *et al.* (2019, *Nature Genetics*). All the additional results have been incorporated in the revised manuscript (lines 150-151, and 156-184). We found that our predicted PAIs were highly enriched in the ChIA-PET loops from the ENCODE (**Fig. 3b**) and PChI-C loops from Jung *et al.* (**Fig. 3c**). More importantly, the fold enrichment increased with the increase of significance level used to claim the PChI-C loops (**Fig. 3c**), consistent with the observation from previous work that Hi-C loops with lower Hi-C *P* values are more reproducible between biological replicates (Jin *et al.*, 2013, *Nature*). Moreover, our SMR & HEIDI method outperformed the correlation-based method using either DNAm or chromatin accessibility data (**Fig. 3c**). We have also shown that our method has similar performance in comparison with the PHM approach. Of the 15,487 causal interactions identified by PHM approach, 10,416 were tested in our SMR & HEIDI analysis; 98.4% were replicated at a nominal significance level ($P_{\text{SMR}} < 0.05$ and $P_{\text{HEIDI}} > 0.01$), and 36% were significant after multiple testing corrections ($P_{\text{SMR}} < 4.8 \times 10^{-6}$ ($0.05/10,416$) and $P_{\text{HEIDI}} > 0.01$). While the PHM approach requires individual-level genotype and chromatin accessibility data and is less computationally efficient due to the use of Bayesian hierarchical model, our SMR & HEIDI method that only requires summary-level data is more flexible and can be potentially applied to all epigenetic QTL data.

We have also shown examples where the predicted PAIs are likely to be functional by integrating GWAS with PAIs (**Fig. 2d** and **Fig. S3**), and these PAIs could be candidates for functional validations in the future. However, we agree with the reviewer that further experimental validations for these novel interactions are needed. We have mentioned this as a limitation of our study in the Discussion (lines 419-421).

Beyond that there are no hard examples shown that differential methylation is recovering known demonstrated E-P interactions.

Re: Our analytical approach was developed to detect the association between DNAm levels of two CpG sites due to the same set of underlying genetic variants rather than detecting differential methylation. We have shown that our predicted PAIs were significantly enriched in the loops identified by different experimental assays (e.g., Hi-C, PCHi-C and ChIA-PET) and further added in the revised manuscript an example that a predicted PAI is validated by an enhancer-promoter (E-P) interaction discovered in two independent Hi-C studies (lines 152-154 and **Fig. S4**).

While the predictions do fall within TADs more frequently by chance, this is not a direct assessment of functional interaction. The enrichment for differentially expressed genes in fig3d looks marginally significant at best.

Re: We agree with the reviewer that the predicted interactions falling within TAD regions were not necessarily functional. However, compared to the chromatin interactions identified by the 3C-based methods, our predicted PAIs may be more relevant to functional interactions, as evidenced by the observation from an additional analysis (see below) that our predicted Pm-PAI genes (genes whose promoters were involved in significant PAI) showed stronger enrichment in active gene groups compared to the predicted target genes from the PCHi-C data (**Fig. S6**). In addition, the use of a genetic model also allows us to integrate PAIs with GWAS results to understand the regulatory mechanisms for complex traits (**Fig. 2d, Fig. 6, Fig. S7, Fig. S10** and lines 236-241).

In the test for enrichment in differentially expressed genes, the Pm-PAI genes were tested against the same number of control genes whose promoter DNAm sites were included in the SMR analysis. This enrichment analysis is conservative because the ascertainment of genes with promoter DNAm sites tested in SMR would potentially lead to upward biases in expression levels of the control genes. Therefore, we performed an additional enrichment analysis by including all the genes available in the GTEx blood samples. We found that most of the control genes (~70%) were in the inactive gene sets and the fold enrichment of our Pm-PAI genes in active gene groups increased substantially (**Fig. S6a**). We also performed a similar enrichment analysis for the predicted target genes from the PCHi-C data in GM12878 cell lines (Jung *et al.*). There was a significant enrichment of the PCHi-C target genes in the active gene groups, but the fold enrichment was slightly smaller than that of our Pm-PAI genes (**Fig. S6b**). We have included these additional results in the revised manuscript (lines 236-241).

Minor:

1. All plots are poorly annotated and difficult to follow. In Fig 1a, where do probe 1 and probe 2 map to on Fig 1b? Also in Fig 1b, are so many acronyms really necessary: PAI, PIDS, PmPmI, PmPAI.

Re: We have tried to improve the figure legends across the manuscript. Specifically, we have swapped Figures 1a and 1b, removed most of the acronyms from Figure 1b (now Figure 1a in the revised manuscript), and labelled the corresponding probes 1 and 2 on it.

2. It should be noted that earlier papers have observed co-varying accessibility (ATAC-seq) and noted that these co-varying peaks could imply direct physical interactions (Kumasaka et al Nature Genetics 2016, Gate et al Nature Genetics 2018, ref 16 is low resolution but these two and Kumasaka et al Nature Genetics 2019 are high resolution) in addition to the Kumasaka et al Nature Genetics 2019 paper.

Re: It is true that co-varying chromatin accessibility has been used to imply the physical interactions (e.g., Gate et al., 2018, *Nature Genetics*). Most of these methods are based on the correlations of epigenomic marks. We have compared the correlation-based method with our SMR & HEIDI method and found that our method outperformed the correlation-based method using either DNA methylation or chromatin accessibility data (**Fig. 3c** and lines 156-184). Our method is flexible and can be applied to other epigenomic data such as chromatin accessibility data from the ATAC-seq. To replicate the interactions identified by Kumasaka *et al.* (2019, *Nature Genetics*) and compare the SMR & HEIDI method with their PHM approach, we applied our method to the summary-level chromatin accessibility QTL (caQTL) data from Kumasaka *et al.* Of the 15,487 causal interactions identified by PHM approach, 10,416 were tested in our SMR & HEIDI analysis; 98.4% were replicated at a nominal significance level ($P_{\text{SMR}} < 0.05$ and $P_{\text{HEIDI}} > 0.01$), and 36% were significant after multiple testing corrections ($P_{\text{SMR}} < 4.8 \times 10^{-6}$ ($0.05/10,416$) and $P_{\text{HEIDI}} > 0.01$). As stated above, while the PHM method from Kumasaka *et al.* requires individual-level genotype and chromatin accessibility data and is less computationally efficient due to the use of Bayesian hierarchical model, our SMR & HEIDI method that requires only summary-level data is more flexible and can be potentially applied to all epigenetic QTL data.

reviewers' Comments:

Reviewer #1:

Remarks to the Author:

Summary:

The authors improved the article, but I still have some important comments.

Major revisions:

1/ The authors wrote in the rebuttal letter: "However, despite the relatively sparse distribution of the predicted PAIs across the genome, the number of predicted PAIs ($m=34,797$) is comparable to the loops identified by experimental assays such as Hi-C and PCHI-C. For example, there are only $\sim 10,000$ Hi-C loops identified from Rao et al. (2014, Cell)..."

This suggests that the predicted PAIs could correspond to loops as defined by Rao et al Cell 2014. Thus, by doing Aggregate Peak Analysis (APA) with a tool like Juicer and Hi-C data, the authors must observe strong enrichment of interactions over local background.

2/ Another way to validate the predicted PAIs is to assess the enrichment of proteins known to be involved in 3D genome including CTCF, Rad21, ZNF143, YY1, Polycomb and PolII. The authors must do such enrichment analysis.

3/ What is the link with allele imbalance of chromatin data? The authors can try to assess the link between the SNPs involved in loops and omics allele imbalance data, for instance using alleleDB, DNA methylation SNPs (<https://science.sciencemag.org/content/361/6409/eaar3146>), H3K4ac/me3/me1 SNPs (<https://science.sciencemag.org/content/364/6439/eaat8266.abstract>).

4/ The authors used Illumina 450K DNAm probes. Could it be possible to impute the other DNAm sites, since this is a common procedure for SNPs in statistical genetics? This would reduce the sparsity effect on the predictions and considerably strengthen the proposed approach.

5/ The authors argued that ROC curves are not suitable to assess their method. Instead the authors used enrichment analysis using PCHI-C data. I think this is an interesting result, but this is not sufficient to properly assess the method.

I think the authors can assess their method in a better way using the following approach, which is related to my 1st comment. For each predicted PAI, the authors can compute a loop calling enrichment value (Hi-C enrichment over background). The authors can use juicer APA to assess the enrichment for a loop (here a PAI) and returns a fold-change and a p-value. I guess the authors can use Fit-Hi-C for the same task, but I am not sure (it would be very interesting to use both since they use very different assumptions to assess enrichment). Then, the authors must compute the correlation (Pearson and Spearman) between the PAI effect size and the loop calling enrichment, and show good correlation. Likewise, they must do the same for the p-values. Using this approach, the authors can compare with the other prediction methods.

Reviewer #2:

Remarks to the Author:

The authors address most of my concerns except the my question #5. At first, why such a "over conservative" test is used such that the fold-enrichment values are very small, which does not make sense. Secondly, why the DNAm pairs within a promoter region were excluded in the PAI analysis? Although the authors added some descriptive statement in line 206-208, there is no clear explanation and rationale.

Reviewer #3:

Remarks to the Author:

While the authors have dutifully compared to several other data sets (PChIC, Rao, ChIA-PET, PHM, Ernst) to support the statistical significance of their predicted interactions from methylation data, the number of loops enriched above significant fold enrichment p-value levels appears to be quite small (the range of a couple hundred loops). I'm not sure whether this means that all these data sets are quite noisy or whether methylation just isn't a good approach for detecting E-P interactions. The significance of what is detectable by this approach just doesn't seem very convincing.

REVIEWERS' COMMENTS

We thank the three reviewers for their additional comments. We have responded to all the additional comments point-by-point below (in blue) and highlighted all the relevant changes (in yellow) in the revised manuscript.

Reviewer #1:

Summary:

The authors improved the article, but I still have some important comments.

Re: We thank the reviewer for acknowledging the improvement of our manuscript.

Major revision:

1. The authors wrote in the rebuttal letter: "However, despite the relatively sparse distribution of the predicted PAIs across the genome, the number of predicted PAIs ($m=34,797$) is comparable to the loops identified by experimental assays such as Hi-C and PChI-C. For example, there are only $\sim 10,000$ Hi-C loops identified from Rao et al. (2014, Cell)..."

This suggests that the predicted PAIs could correspond to loops as defined by Rao et al Cell 2014. Thus, by doing Aggregate Peak Analysis (APA) with a tool like Juicer and Hi-C data, the authors must observe strong enrichment of interactions over local background.

Re: As suggested by the reviewer, we have performed an Aggregate Peak Analysis (APA) using Juicer (Durand *et al.*, 2016, Cell systems) and observed a strong aggregate enrichment of our predicted PAIs in the combined Hi-C map from GM12878 from Rao *et al.* (2014, Cell) (APA score = 7.63, Z-score = 50.7; **Table S1**). We have added this result in the revised manuscript (lines 184-189).

2. Another way to validate the predicted PAIs is to assess the enrichment of proteins known to be involved in 3D genome including CTCF, Rad21, ZNF143, YY1, Polycomb and PolIII. The authors must do such enrichment analysis.

Re: We thank the reviewer for this suggestion. We have tested whether our predicted PAIs were enriched in the binding regions of proteins known to be involved in 3D organization of the genome. We used the chromatin immuno-precipitation sequencing (ChIP-Seq) data from GM12878 for four DNA-binding proteins (i.e., CTCF, Rad21, ZNF143, YY1) from the ENCODE project. Of the 21,787 unique DNAm sites that showed significant PAIs, 9,454 (43.4%), 7,588 (34.8%), 6,854 (31.5%), and 9,477 (43.5%) were located in the binding regions of CTCF, Rad21, ZNF143, and YY1, respectively (**Methods**). These overlaps were significantly larger than those for a random set of DNAm sites tested in the PAI analysis (1.14-fold on average, $P < 0.001$) or a random set of genomic sites (3.81-fold on average, $P < 0.001$) (**Fig. S7a**; lines 219-227).

3. What is the link with allele imbalance of chromatin data? The authors can try to assess the link between the SNPs involved in loops and omics allele imbalance data, for instance using

alleleDB, DNA methylation SNPs

(<https://science.sciencemag.org/content/361/6409/eaar3146>), H3K4ac/me3/me1 SNPs (<https://science.sciencemag.org/content/364/6439/eaat8266.abstract>).

Re: We have tested whether the top associated mQTLs of the DNAm sites that showed significant PAIs were enriched for variants associated with allele-specific DNAm identified from Onuchic *et al.* (Science, 2018). There were 385 PAI mQTLs overlapping with variants associated with allele-specific DNAm, and the overlap was significantly larger than that of the same number of mQTLs randomly sampled from the top associated mQTLs of all the DNAm sites used in the PAI analysis (1.44-fold, $P < 0.001$) (**Fig. S7b**; lines 229-235).

4. The authors used Illumina 450K DNAm probes. Could it be possible to impute the other DNAm sites, since this is a common procedure for SNPs in statistical genetics? This would reduce the sparsity effect on the predictions and considerably strengthen the proposed approach.

Re: We agree with the reviewer that imputation may help to reduce the sparsity of the predicted PAIs. However, because imputation requires individual-level DNAm data while our study uses only summary-level DNAm data, it is unfeasible to perform imputation in this study. Moreover, although several methods have been proposed to impute DNAm data, the imputation accuracy is still far from what can be achieved in SNP imputation (Zou *et al.*, 2018, BMC Genomics). We have commented on this in the revised manuscript (lines 437-439).

5. The authors argued that ROC curves are not suitable to assess their method. Instead the authors used enrichment analysis using PCHi-C data. I think this is an interesting result, but this is not sufficient to properly assess the method. I think the authors can assess their method in a better way using the following approach, which is related to my 1st comment. For each predicted PAI, the authors can compute a loop calling enrichment value (Hi-C enrichment over background). The authors can use juicer APA to assess the enrichment for a loop (here a PAI) and returns a fold-change and a p-value. I guess the authors can use Fit-Hi-C for the same task, but I am not sure (it would be very interesting to use both since they use very different assumptions to assess enrichment). Then, the authors must compute the correlation (Pearson and Spearman) between the PAI effect size and the loop calling enrichment, and show good correlation. Likewise, they must do the same for the p-values. Using this approach, the authors can compare with the other prediction methods.

Re: As suggested by the reviewer, we have performed an Aggregate Peak Analysis (APA) to examine the overlap of the predicted interactions with the combined Hi-C map in GM12878 from Rao *et al.* (Cell, 2014). Since the APA approach was developed to test the aggregate enrichment of a set of putative peaks rather than each peak individually (Rao *et al.*, Cell, 2014), we evaluated the performance of different interaction prediction methods (i.e., correlation, PHM, and SMR & HEIDI) by the aggregate enrichment of the predicted interactions in the Hi-C map using Juicer. The results showed that the predicted interactions from all the methods were enriched in the combined Hi-C map from the Rao *et al.* study, and that our predicted PAIs from DNAm data showed the strongest enrichment among all the predicted interactions although the

APA score of PHM was higher than that of SMR & HEIDI using chromatin accessibility data (**Table S1**; lines 184-189).

Reviewer #2

(Remarks to the Author):

The authors address most of my concerns except the my question #5. At first, why such a "over conservative" test is used such that the fold-enrichment values are very small, which does not make sense. Secondly, why the DNAm pairs within a promoter region were excluded in the PAI analysis? Although the authors added some descriptive statement in line 206-208, there is no clear explanation and rationale.

Re: We used the conservative test to avoid the ascertainment bias introduced by the use of DNAm array data. Note that the DNAm probes on an Illumina 450K methylation array are not random but designed to target certain functional elements such as promoters (**Fig. 4a**). Therefore, the "outcome" DNAm probes assessed in the PAI analysis are functionally enriched regardless of whether they are involved in interactions. Our enrichment test is to assess whether the promoter-interacting DNAm sites are more functionally enriched than the array sites, which explains why most of the fold enrichment values were small. Nevertheless, we have now also reported the fold enrichment values computed against random genomic positions wherever appropriate (**Fig. S5b**), which were several-fold larger than those computed against array probes. We have clarified this in the revised manuscript (lines 205-213).

The reasons why we excluded the DNAm pairs within a promoter region are because the chromatin interactions identified from Hi-C are often between a promoter region and nearby regions (i.e., the interactions within a promoter region are not studied) and because it helps reduce the computational and multiple testing burdens. We have clarified this in the revised manuscript (lines 472-476).

Reviewer #3

(Remarks to the Author):

While the authors have dutifully compared to several other data sets (PChIC, Rao, ChIA-PET, PHM, Ernst) to support the statistical significance of their predicted interactions from methylation data, the number of loops enriched above significant fold enrichment p-value levels appears to be quite small (the range of a couple hundred loops). I'm not sure whether this means that all these data sets are quite noisy or whether methylation just isn't a good approach for detecting E-P interactions. The significance of what is detectable by this approach just doesn't seem very convincing.

Re: We thank the reviewer for the additional comment.

Regarding the small fold enrichment values, we have clarified in the manuscript (lines 205-213) that it was owing to the use of a conservative test, i.e., testing whether the promoter-interacting

DNAm sites are more functionally enriched than control probes randomly sampled from all the “outcome” probes used in the PAI analysis. The outcome probes, however, were ascertained because the DNAm probes on an Illumina 450K methylation array are designed to target certain functional elements such as promoters (**Fig. 4a**). So, the test is essentially to assess whether the promoter-interacting DNAm sites are more functionally enriched than the array sites, which explains why most of the fold enrichment values were small. We have now also reported the fold enrichment values computed against random genomic positions wherever appropriate, which were several-fold larger than those computed against array probes (**Fig. S5b**; lines 211-213).

We agree with the reviewer that although most PAIs are within TADs, the overlaps of PAIs with Hi-C loops are limited. We have commented on this issue in the Discussion section (also see below).

“There are several reasons why the overlaps between the predicted PAIs and Hi-C loops were limited. **First and most importantly, Hi-C loops were detected with substantial noises. We observed that the concordances between different Hi-C data sets were very limited (Fig. S13)**, consistent with the conclusion from Forcato et al. that the reproducibility of Hi-C loops is low at all resolutions. Second, most (65%) of our predicted PAIs are interactions between DNAm sites within 50 Kb (Fig. S2b), which are often not well captured by the 3C-based methods due to its low resolution. Third, the chromatin interactions are cell type specific so that differences between the Hi-C loops identified in cell lines and our PAIs identified in whole blood are expected. For the PAIs that were between DNAm sites not located in TADs or Hi-C loops, we have shown specific examples that these predicted PAIs are likely to be functionally interacted (Fig. 2d and Fig. S3), suggesting that these PAIs are likely to be interactions yet to be identified by experimental assays. On the other hand, compared to the loops identified based on 3C-based methods, our predicted PAIs are more likely to be functional interactions due to the use of genetic and regulatory epigenomic data, as evidenced by the observation that our predicted Pm-PAI genes showed stronger enrichment in active gene groups compared to the predicted target genes from the PCHi-C data (Fig. S8).”

Reviewers' Comments:

Reviewer #1:

Remarks to the Author:

Summary:

The authors significantly improved the article, but I still have one important comment.

Major revision:

In my previous comment 5, I suggested to compute the correlation (Pearson and Spearman) between the PAI effect size and the loop calling enrichment, and show good correlation. Why? Because APA enrichment results are not enough to evaluate the accuracy of predictions. Indeed APA enrichment depends for instance on the number of loops an algorithm calls. For instance, if I have a very stringent threshold for my loop prediction method, then it's more likely to have a high enrichment compared to the other methods. I suggested to the authors to use Juicer APA software for this task. APA is designed for aggregate enrichment, but it can also be easily used to access to the loop enrichment (for every loop) by using intermediate files computed by APA software. The authors can use for instance the file enhancement.txt (see APA doc "The enhancement.txt contains a list of the P2M values for each of the standard APA submatrices (one value per putative loop)"). I thus suggest to compute the correlation between the PAI effect sizes and P2M values and to compare with the other loop prediction methods.

Reviewer #2:

Remarks to the Author:

My questions have been addressed.

REVIEWERS' COMMENTS

We have responded to the additional comment from Reviewer #1 below and have highlighted the relevant change (in yellow) in the revised manuscript.

Reviewer #1:

(Remarks to the Author):

Summary:

The authors improved the article, but I still have some important comments.

Re: We thank the reviewer for acknowledging the improvement of our manuscript.

Major revision:

1. In my previous comment 5, I suggested to compute the correlation (Pearson and Spearman) between the PAI effect size and the loop calling enrichment, and show good correlation. Why? Because APA enrichment results are not enough to evaluate the accuracy of predictions. Indeed APA enrichment depends for instance on the number of loops an algorithm calls. For instance, if I have a very stringent threshold for my loop prediction method, then it's more likely to have a high enrichment compared to the other methods. I suggested to the authors to use Juicer APA software for this task. APA is designed for aggregate enrichment, but it can also be easily used to access to the loop enrichment (for every loop) by using intermediate files computed by APA software. The authors can use for instance the file enhancement.txt (see APA doc "The enhancement.txt contains a list of the P2M values for each of the standard APA submatrices (one value per putative loop)"). I thus suggest to compute the correlation between the PAI effect sizes and P2M values and to compare with the other loop prediction methods.

Re: We compared the prediction methods using only the aggregate enrichment value because the Aggregate Peak Analysis (APA) is developed to test the aggregate enrichment of a set of putative peaks in a Hi-C map and because the aggregate enrichment value rather than the loop enrichment value is recommended to summarize the result in the Juicer APA document.

As far as we understand, the underlying hypothesis of this comment is that the strength of a Hi-C loop (or predicted interaction) should correlate with the APA enrichment value. To test this hypothesis, we used the 9,448 high quality Hi-C loops from Rao *et al.* (2014, *Cell*) as a benchmark. We first performed the APA using Juicer (Durand *et al.*, 2016, *Cell* systems) to compute the loop enrichment value for each of the 9,448 Hi-C loops in the combined Hi-C map from Rao *et al.* (2014, *Cell*). We then correlated the strength of the Hi-C loops with the APA loop enrichment values and observed no correlation ($r = -0.0086$) between them.

We also correlated the strength of predicted interaction (i.e., $-\log_{10}(P\text{-value})$) from each of the three prediction methods (i.e., correlation-based method, pairwise hierarchical model (PHM), and SMR & HEIDI) with the APA loop enrichment value, and again observed no correlation for all the prediction methods (see **Table R1** below).

We have added this result in the revised manuscript (**Table S2**, lines 190-193).

Table R1 Pearson correlation between the strength of a loop or predicted interaction (i.e., $-\log_{10}(P\text{-value})$) and the APA loop enrichment value (i.e., Juicer P2M value).

Interaction prediction method	Hi-C Map	Pearson correlation
9,448 Hi-C loops identified from Rao et al.	In_situ Hi-C GM12878	-0.0086
SMR&HEIDI using DNAm	In_situ Hi-C GM12878	-0.0219
SMR&HEIDI using ATAC-seq	In_situ Hi-C GM12878	0.0030
PHM using ATAC-seq	In_situ Hi-C GM12878	-0.0016
Correlation-based method using DNAm	In_situ Hi-C GM12878	0.0002
Correlation-based method using ATAC-seq	In_situ Hi-C GM12878	0.0031
A control set	In_situ Hi-C GM12878	0.0003

Reviewer #2:

(Remarks to the Author):

My questions have been addressed.

Re: We thank the reviewer for the comments on all versions of our manuscript.

Reviewers' Comments:

Reviewer #1:

Remarks to the Author:

I am satisfied that the authors have addressed all my comments.

REVIEWERS' COMMENTS

Reviewer #1:

I am satisfied that the authors have addressed all my comments.

Re: We thank the reviewer for the comments on all versions of our manuscript.